# The level of oncogenic Ras determines the malignant transformation of Lkb1 mutant tissue in vivo

Briana Rackley[1,2,8], Chang-Soo Seong[1,8], Evan Kiely [1,7], Rebecca E. Parker [1,2], Manali Rupji [3], Bhakti Dwivedi[4], John M. Heddleston [5], William Giang [6], Neil Anthony[6], Teng-Leong Chew [5] & Melissa Gilbert-Ross [1✉]

The genetic and metabolic heterogeneity of RAS-driven cancers has confounded therapeutic strategies in the clinic. To address this, rapid and genetically tractable animal models are needed that recapitulate the heterogeneity of RAS-driven cancers in vivo. Here, we generate a *Drosophila melanogaster* model of Ras/*Lkb1* mutant carcinoma. We show that low-level expression of oncogenic Ras (Ras$^{Low}$) promotes the survival of *Lkb1* mutant tissue, but results in autonomous cell cycle arrest and non-autonomous overgrowth of wild-type tissue. In contrast, high-level expression of oncogenic Ras (Ras$^{High}$) transforms *Lkb1* mutant tissue resulting in lethal malignant tumors. Using simultaneous multiview light-sheet microcopy, we have characterized invasion phenotypes of Ras/*Lkb1* tumors in living larvae. Our molecular analysis reveals sustained activation of the AMPK pathway in malignant Ras/*Lkb1* tumors, and demonstrate the genetic and pharmacologic dependence of these tumors on CaMK-activated Ampk. We further show that *LKB1* mutant human lung adenocarcinoma patients with high levels of oncogenic KRAS exhibit worse overall survival and increased AMPK activation. Our results suggest that high levels of oncogenic KRAS is a driving event in the malignant transformation of *LKB1* mutant tissue, and uncovers a vulnerability that may be used to target this aggressive genetic subset of RAS-driven tumors.

[1] Department of Hematology and Medical Oncology, Emory University School of Medicine, Atlanta, GA, USA. [2] Cancer Biology Graduate Program, Emory University, Atlanta, GA, USA. [3] Biostatistics Shared Resource, Winship Cancer Institute of Emory University, Atlanta, GA, USA. [4] Bioinformatics and Systems Biology Shared Resource, Winship Cancer Institute of Emory University, Atlanta, GA, USA. [5] Advanced Imaging Center, Janelia Research Campus, Howard Hughes Medical Institute, Ashburn, VA, USA. [6] Integrated Cellular Imaging Core, Emory University School of Medicine, Emory University, Atlanta, GA, USA. [7] Present address: Winship Research Informatics, Winship Cancer Institute of Emory University, Atlanta, GA, USA. [8] These authors contributed equally: Briana Rackley, Chang-Soo Seong. ✉email: mmgilbe@emory.edu

KRAS is the most commonly mutated oncogene in human cancer, and is frequently mutated in cancer types associated with high mortality such as non-small cell lung cancer. Efforts to directly target the KRAS protein have been challenging, although renewed efforts are currently in clinical trials[1]. Large-scale sequencing of lung adenocarcinoma has uncovered heterogeneity in mutant KRAS tumors due to concomitantly mutated tumor suppressor genes such as *TP53* and *LKB1*, genetic subtypes that are largely mutually exclusive and which harbor distinct biologies and therapeutic susceptibilities[2]. An added layer of complexity arises due to the extensive metabolic rewiring observed in RAS-driven tumors[3], which can arise due to Kras-mutant dosage and alterations in signaling pathways downstream of mutated tumor suppressor genes[4]. Increasingly, metabolic rewiring is known to be dependent on tissue-level dynamics within the tumor and the tumor microenvironment. Therefore, there is a need to develop rapid and powerful models of RAS-driven cancers that mimic the complex landscape of these tumors in vivo.

Liver Kinase B1 (LKB1) is a master serine/threonine kinase that phosphorylates 13 downstream kinases of the AMP-activated protein kinase family (AMPK) family to control cell growth and cell polarity[5]. LKB1 activity is lost in a wide spectrum of human cancers and the gene that encodes LKB1 (*STK11*) is the third most frequently mutated tumor suppressor in human lung adenocarcinoma. Loss of *LKB1* frequently occurs in KRAS-driven lung adenocarcinoma, and has been shown to promote metastasis, shorten overall survival, and confer resistance to targeted therapies and checkpoint inhibitors[6–10]. Altogether, these differences in survival and treatment outcomes highlight the importance of in vivo models that recapitulate the complexity and heterogeneity of these tumors when developing and implementing cancer treatments.

*Drosophila melanogaster* is a powerful model system for studying cancer biology due to the high conservation of human oncogene and tumor suppressor pathways[11,12]. Elegant genetic mosaic techniques in *Drosophila* allow tissue-specific overexpression of oncogenes and knockdown of tumor suppressors within distinct subpopulations of cells, which bestows the ability to build complex tumor landscapes in vivo. Seminal work using these methods has identified cooperating mutations that promote the metastasis of benign *Kras*-mutant tumors in vivo, and has identified such cooperating models as amenable to pharmacologic approaches[13–16]. However, despite evidence from mouse models that loss of *Lkb1* is sufficient to promote tumor progression and metastasis in *Kras*-mutant lung tumors[17], there has been no report of malignant synergy between *Ras* and *Lkb1* using the rapid and genetically tractable *Drosophila* model.

Here, using a *Drosophila* model of *Ras/Lkb1*-driven malignant progression, we found that the relative levels of oncogenic Ras determine clonal growth dynamics in *Lkb1* mutant tissue. Low levels of oncogenic Ras promote non-autonomous growth of surrounding wild-type tissue, while high levels promote malignant progression and organismal lethality. To further characterize the metastatic capability of *Ras/Lkb1* malignant cells we used simultaneous multiview light-sheet microscopy to image live tumor-bearing larvae for up to 48 h, and show that *Ras/Lkb1* cells actively degrade basement membrane, and ultimately invade distant tissues. To further define the mechanism driving the progressive synergy between high oncogenic Ras and loss of *Lkb1* we investigated signaling networks in mosaic tissue. We show that malignant *Ras/Lkb1* tumors activate AMPK and are dependent on the activation of the *Drosophila* ortholog of CAMKK2. We validate the translational potential of our work by showing high-level KRAS with concurrent mutation in *LKB1* represents a unique subset of patients with worse overall survival and increased AMPK activation. Our work uncovers oncogenic *KRAS* copy number gains or amplification as a synergistic mechanism that drives the aggressive nature of *LKB1* mutant tumors. In addition, our work proves *Drosophila* as a powerful model for the rational design of targeted therapies for genetic subsets of RAS-driven cancers, and suggests that the *LKB1* subset of KRAS-driven cancers may benefit from targeting of the CAMKK/AMPK circuit.

## Results

**Clonal loss of *Lkb1* in vivo results in autonomous cell death.**
Recent work has highlighted effects of the dosage of oncogenic Ras on the progression of Ras-dependent cancers[18,19]. Previous work in *Drosophila* has identified myriad pathways that collaborate with mutant Ras to promote tumor progression and metastasis[20], but how the dosage of Ras affects tumor progression in these multiple hit models is unknown. To address this question, we identified oncogenic Ras transgenes with differing expression levels. One expresses oncogenic Ras at levels similar to endogenous Ras (Ras$^{Low}$). The other expresses Ras at levels several-fold higher (Ras$^{High}$) (Fig. 1b and Supplementary Fig. 4). To mimic the genetic landscape of human KRAS-driven cancers we chose to co-mutate the tumor suppressor *LKB1* in Ras$^{Low}$ and Ras$^{High}$ tissue. Most tumor-specific *LKB1* mutations are homozygous deletions or loss-of-heterozygosity with somatic mutation[21–23]. Among the latter, nonsense or frameshift mutations leading to protein truncation are the most common[24]. To identify the *Drosophila Lkb1* loss-of-function allele with the strongest reduction in Lkb1 protein levels we first generated an antibody to *Drosophila* Lkb1. We then assayed for Lkb1 protein in transheterozygous larvae using three previously published *Lkb1* loss-of-function alleles (X5[25], 4B1-11, and 4A4-2[26]) over a large deletion that removes the *Lkb1* gene. The *Lkb1*$^{X5}$ and *Lkb1*$^{4B1-11}$ loss-of-function alleles reduced Lkb1 protein expression by 60% compared to control. However, the *Lkb1*$^{4A4-2}$ allele reduced protein expression by 80% (Fig. 1a and Supplementary Fig. 4), which agrees with prior published genetic data suggesting *Lkb1*$^{4B1-11}$ as having residual protein activity[27]. The *Lkb1*$^{4A4-2}$ allele was chosen for further study and will be referred to as *Lkb1*$^{-/-}$.

We used the GFP-labeled eye expression system[14] to express Ras$^{Low}$ in discreet patches or 'clones' of developing eye epithelial tissue. Expression of Ras$^{Low}$ resulted in ablation of eye tissue and benign outgrowths of eye cuticle similar to what has been reported in prior reports using a UAS-Ras$^{V12}$ transgene[14,28] (Fig. 1c). We then used the GFP-labeled eye expression system to inactivate the *Lkb1* tumor suppressor (*Lkb1*$^{-/-}$) in clones of cells in the developing eye. Inactivation of *Lkb1* in clones resulted in adult flies with small, rough eyes (Fig. 1c), suggesting high levels of apoptosis. To test this, we assayed for cleaved death caspase 1 (DCP1) in mutant clones using immunofluorescence in wandering 3rd instar eye-imaginal discs. As expected, loss of *Lkb1* (marked by GFP+ tissue) resulted in a large increase in autonomous cleaved DCP1 expression as compared to discs carrying control FRT82B clones (Fig. 1d, e and Supplementary Data 1). These data suggest that homozygous loss of *Lkb1* within an otherwise wild-type epithelium can result in a high level of apoptosis in vivo.

**Low-level Ras and loss of *Lkb1* synergize to promote non-autonomous benign overgrowth.** Data from genetically engineered mouse models suggests loss of *Lkb1* is sufficient to promote the progression and metastasis of nascent Kras-mutant lung adenocarcinoma[17]. Due to the redundancy of the vertebrate genome and paucity of rapid genetic mosaic analyses

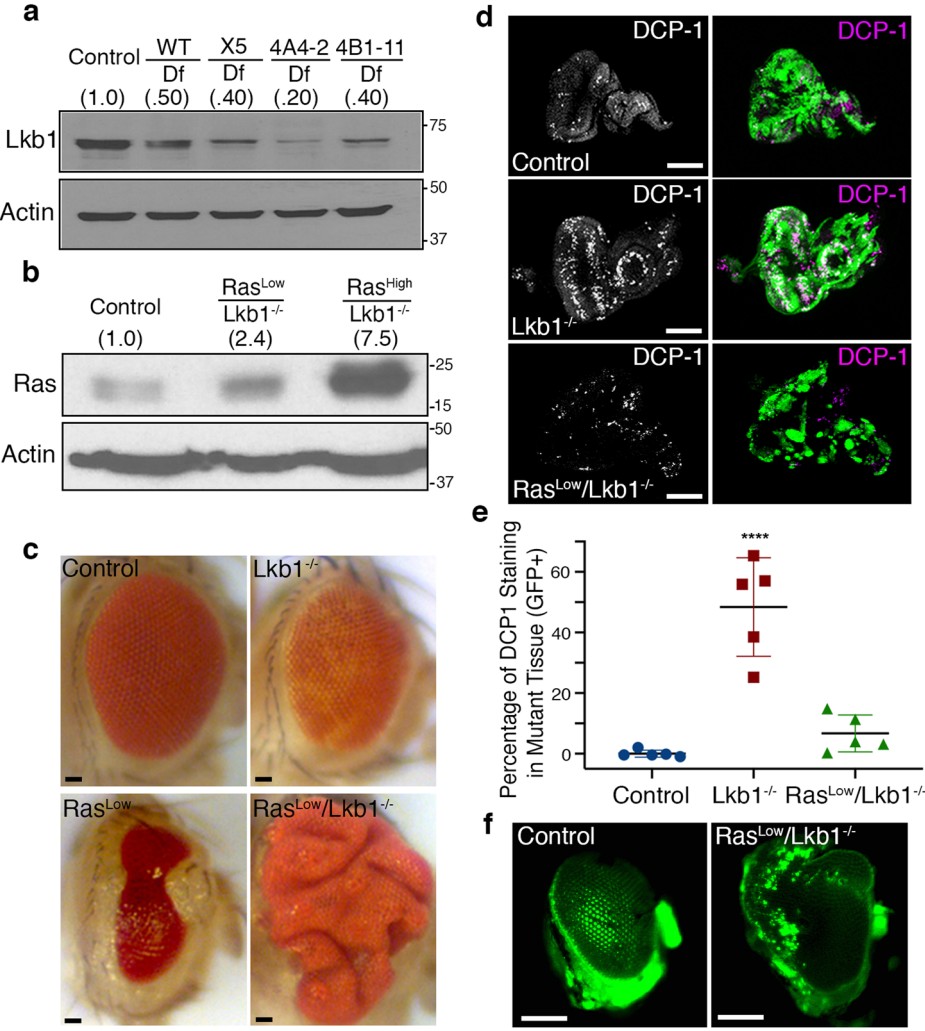

**Fig. 1 Clonal loss of *Lkb1* in vivo results in autonomous cell death. a** Western analysis of Lkb1 protein in larvae transheterozygous for a deletion (*Df(3R) Exel6169*) that removes the *Lkb1* gene, and either a wild-type third chromosome or three loss-of-function alleles of *Lkb1* (X5, 4A4-2, 4B1-11). **b** Western analysis of Ras levels in mosaic eye-imaginal disks from the indicated genotypes (control = FRT82B). Note the Ras antibody detects both endogenous and oncogenic Ras. **c** Representative brightfield images of mosaic adult eyes with clones of the indicated genotypes. Scale bar, 20 μm. **d** Confocal maximum intensity projections of third instar mosaic eye discs carrying GFP-tagged clones of the indicated genotypes, and stained for endogenous death caspase 1 (DCP1, magenta). Scale bar, 100 μm. **e** Percentage of DCP1 staining in GFP-positive mutant tissue was quantified from $n = 5$ imaginal discs per condition using thresholding in FIJI (ImageJ). Data are represented as mean with error bars representing standard deviation (****$p = 0.000006$, one-way ANOVA with multiple comparisons). **f** Fluorescent images of adult eyes carrying GFP-labeled clones of the indicated genotypes. Images are representative of $n = 10$ independent flies per genotype. Scale bar, 100 μm.

in mice, we sought to use the GFP-labeled *Drosophila* eye expression system to build a Ras/*Lkb1* model of cooperative tumorigenesis. We simultaneously expressed Ras$^{Low}$ and depleted *Lkb1* (Ras$^{Low}$/*Lkb1*$^{-/-}$) in clones of developing eye epithelial tissue, and found that autonomous DCP1 levels returned to those observed in control eye-imaginal disks (Fig. 1e; Δ mean = +41.7 [95% CI, 26.0–57.4]. These data suggest that low levels of oncogenic Ras promote the survival of *Lkb1*$^{-/-}$ mutant tissue in vivo. In addition, eye-imaginal disc complexes carrying Ras$^{Low}$/*Lkb1*$^{-/-}$ clones were larger than mosaic control discs but contained only a small amount of mutant GFP+ tissue compared to the expression of Ras$^{Low}$ alone. In agreement with these results, analysis of adult Ras$^{Low}$/*Lkb1*$^{-/-}$ mosaic eyes revealed a large, overgrown eye phenotype composed of mostly GFP− wild-type cells (Fig. 1c, f). To confirm the overgrown eye phenotype was due to synergy between Ras and *Lkb1* and not to simply preventing cell death in *Lkb1* mutant cells we expressed the baculoviral caspase

inhibitor p35 in *Lkb1* mutant clones. Expressing p35 in *Lkb1* mutant clones resulted in a majority of flies with eyes that are phenotypically similar to expression of p35 alone (normal size eye), with 20% of flies exhibiting a more severe smaller malformed eye (Supplementary Fig. 1).

To investigate the mechanism that results in an increase in organ size in Ras$^{Low}$/*Lkb1*$^{-/-}$ flies, we analyzed BrdU incorporation in mosaic Ras$^{Low}$/*Lkb1*$^{-/-}$ eye-imaginal disc tissue. Eye disc tissue carrying Ras$^{Low}$/*Lkb1*$^{-/-}$ clones exhibits BrdU incorporation in GFP− wild-type cells surrounding mutant clones (Fig. 2a). In addition, we analyzed mosaic Ras$^{Low}$/*Lkb1*$^{-/-}$ eye-imaginal disc tissue by fluorescence-activated cell sorting (FACS). This analysis revealed an increase in the percentage of GFP+ mutant cells in G1 when compared to GFP+ cells from control FRT82B discs (Fig. 2b, c and Supplementary Data 2). Altogether, these data suggest that although Ras$^{Low}$/*Lkb1*$^{-/-}$ mutant cells survive, they undergo G1 arrest while promoting the increased hyperplastic proliferation of surrounding wild-type tissue.

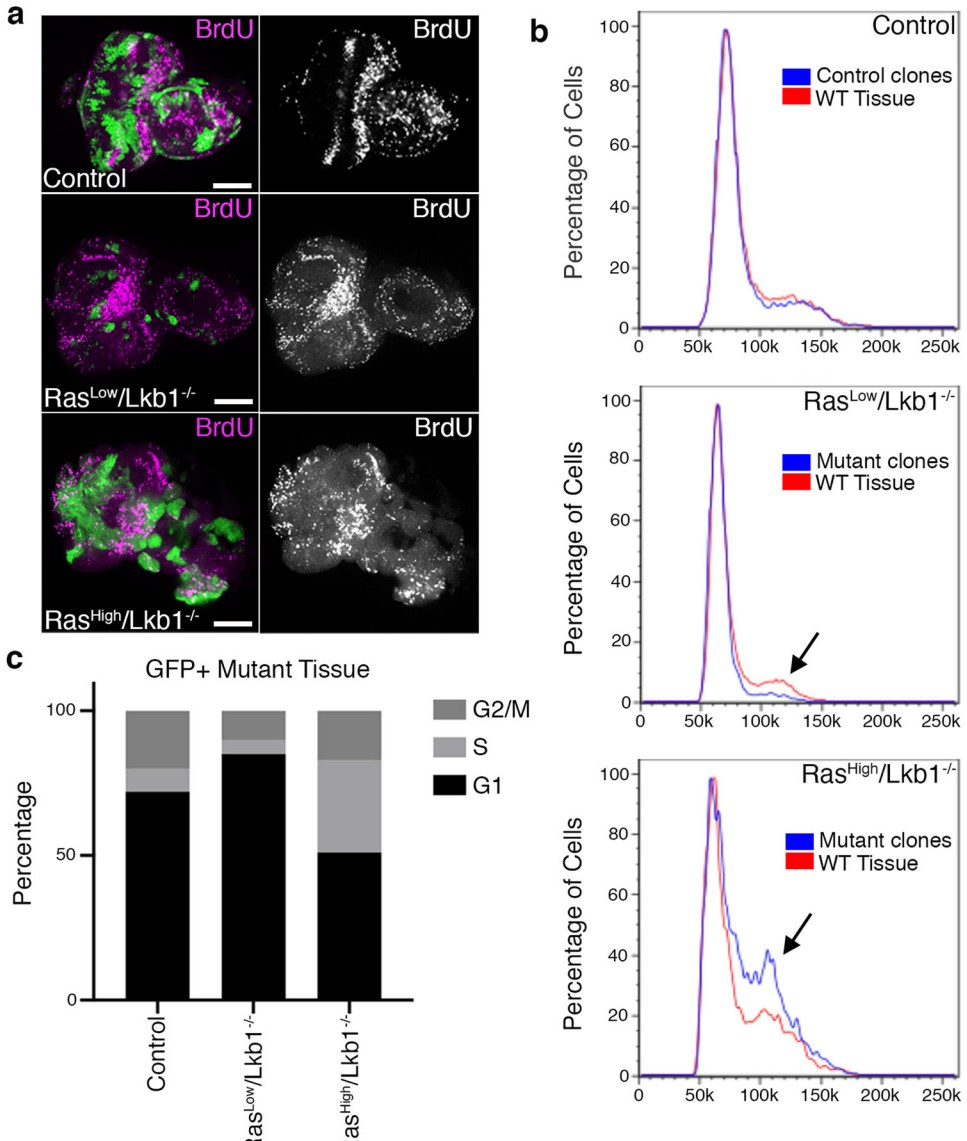

**Fig. 2 The level of oncogenic Ras determines distinct autonomous vs. non-autonomous cell-cycle phenotypes in *Lkb1* mutant tissue. a** Confocal images of mosaic eye-imaginal discs carrying GFP+ clones of the indicated genotypes (control = *FRT82B*), and stained for BrdU incorporation (magenta). Images are representative of *n* = 10 independent eye-imaginal discs per genotype. Scale bar, 100 μm. **b** Fluorescence-activated cell sorting (FACS) analysis of mosaic eye-imaginal discs with GFP-labeled clones of the indicated genotypes. Black arrows point to shifts in relative cell-cycle phasing. Analysis is representative of *n* = 3 independent experiments of 20 mutant imaginal discs/genotype and 40 imaginal discs/genotype for control. **c** Histogram showing percentage of GFP-labeled control or mutant cells in each phase of the cell cycle.

**High-level oncogenic Kras promotes the neoplastic transformation of *Lkb1* mutant tissue.** Previous studies have implicated the dose of mutant Kras in tumor progression, cell motility, and metabolic reprogramming[18,19,29], therefore we used the GFP-labeled eye expression system to clonally express Ras[High] and mutate *Lkb1* in developing eye epithelia (Ras[High]/*Lkb1*[−/−]). When combined with *Lkb1* loss-of-function, expression of Ras[High] resulted in severely overgrown and disorganized 3rd instar larval eye-imaginal disc tumors composed of mostly GFP+ mutant tissue (Fig. 3a). FACS analysis of mutant tissue revealed a shift in cell-cycle phasing that favored G2/M, suggesting that mutant cells were precociously completing G1 (Fig. 2b, c). The majority of larvae carrying Ras[High]/*Lkb1*[−/−] mosaic discs did not pupate but continued to grow into 'giant larvae' while expression of Ras[High] alone resulted in late pupal lethality (Fig. 3b). The giant larval phenotype is shared by loss-of-function mutations in the *Drosophila* neoplastic tumor suppressor genes[30] and suggests that

Ras[High]/*Lkb1*[−/−] tumors are malignant. To test this, we performed an allograft assay by implanting control, Ras[Low]/*Lkb1*[−/−], and Ras[High]/*Lkb1*[−/−] GFP+ tumor tissue in the abdomens of wild-type hosts. Transplanted control and Ras[Low]/*Lkb1*[−/−] tissue failed to grow in host abdomens (Fig. 3e). Surprisingly, the lifespan of hosts with transplanted Ras[Low]/*Lkb1*[−/−] tissue was shortened which suggests that residual GFP− 'wild-type' tissue from the transplant could be partially transformed. In contrast, only transplanted Ras[High]/*Lkb1*[−/−] tissue was able to grow into visible secondary tumors that significantly shortened host survival (Fig. 3e, f and Supplementary Data 3) thus confirming the malignancy of Ras[High]/*Lkb1*[−/−] tumor tissue.

**High-level Ras promotes the invasion and metastasis of Lkb1 mutant tissue.** Mutations in cell polarity proteins cooperate with oncogenic Ras to drive tumor cell invasion and metastasis[20].

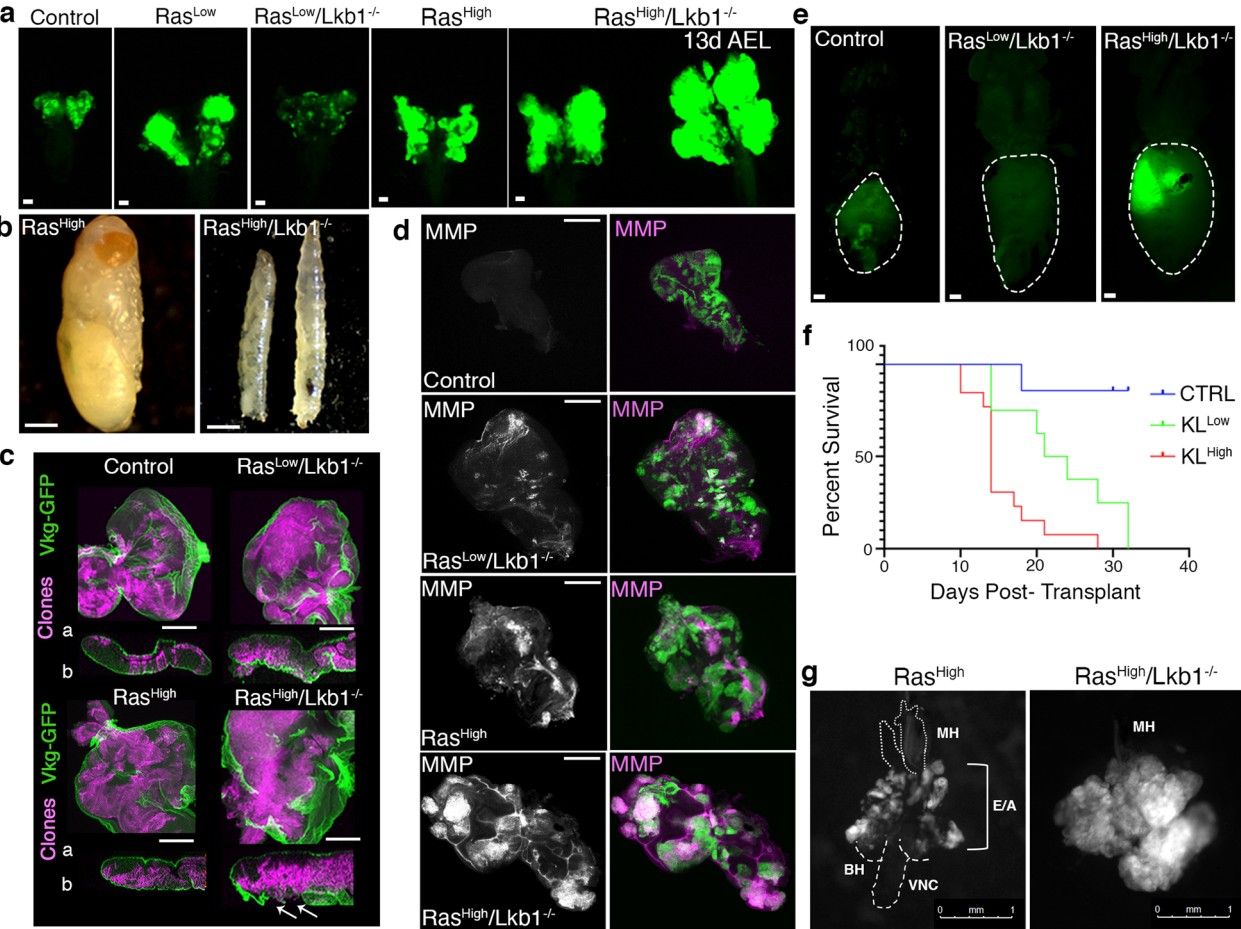

**Fig. 3 Oncogenic Ras^High promotes the malignant transformation of *Lkb1* mutant tissue. a** Fluorescent images of 3rd instar larval eye-imaginal discs (exception labeled '13d AEL') carrying GFP+ clones of the indicated genotypes (control = FRT82B). Images are representative of n = 10 independent eye-imaginal discs per genotype. Scale bar, 20 μm. AEL = after egg-lay. The stage '13 days AEL' is indicative of a larva that failed to pupate at day ~5d AEL, and is a classic neoplastic phenotype. **b** Representative brightfield image of the lethal stage of a fly carrying Ras^High clones (left) and Ras^High/*Lkb1*^−/− clones (right). Note that both age-matched third instar and giant larvae are shown for the Ras^High/*Lkb1*^−/− genotype. **c** Confocal images of eye-imaginal discs carrying RFP^+ clones (magenta) of the indicated genotypes and expressing type IV collagen-GFP (Vkg-GFP). White arrow indicates breaks in Vkg-GFP. a = apical, b = basal. Images are representative of n = 5 independent eye-imaginal discs per genotype. Scale bar, 100 μm. **d** Confocal images of third instar eye discs carrying GFP+ clones of the indicated genotypes, and stained for matrix metalloproteinase 1 (MMP1, magenta in overlay). Images are representative of n = 10 independent eye-imaginal discs per genotype. Scale bar, 100 μm. **e** Fluorescent images of w^1118 adult virgin female hosts carrying transplanted allografts of 3rd instar eye-imaginal discs with GFP+ clones of the indicated genotypes. Scale bar, 100 μm. **f** Quantification of survival post-transplant in allograft assay. Survival was measured from 7 days post-transplant to time of death. CTRL (FRT82B, n = 7), KL^Low (Ras^Low/*Lkb1*^−/−, n = 8), and KL^High (Ras^High/*Lkb1*^−/−, n = 13) (CTRL-KL^Low, **p = 0.0030, CTRL-KL^High, ***p = 0.0001, KL^Low- KL^High, *p = 0.0129, Log-rank test).
**g** Representative fluorescent images of dissected cephalic complexes and ventral nerve cord (VNC) from larvae carrying GFP^+ clones (white) of the indicated genotypes. BH = brain hemispheres; E/A = eye/antennal discs, MH = mouth hooks. The Ras^High/*Lkb1*^−/− tissue completely invades and obscures contiguous organs. Scale bar, 1 mm. Images are representative of n = 10 cephalic complexes/genotype.

Previous studies have shown that Lkb1 regulates cell polarity and epithelial integrity across species[31,32], therefore, we hypothesized that malignant Ras^High/*Lkb1*^−/− tumors would have invasive properties. To test this, we first examined whether Ras/*Lkb1* mutant cells compromised basement membrane structure by examining the expression of GFP-tagged Collagen IV (Viking (Vkg)-GFP) using conventional fixation and confocal microscopy. Compared to control and Ras^Low/*Lkb1*^−/− tissue which shows contiguous Vkg-GFP expression in epithelia, Ras^High/*Lkb1*^−/− tissue exhibits breaks in Vkg-GFP expression (Fig. 3c). Expressing Ras^High on its own is lethal (albeit at the pharate adult stage), so we investigated Vkg-GFP in this genotype and once again found no breaks in the structure of Vkg-GFP. We next assayed matrix metalloproteinase (MMP) expression, as MMPs degrade basement membrane. Compared to control, Ras^Low/*Lkb1*^−/−, and Ras^High clones, Ras^High/*Lkb1*^−/− mutant

tissues express high levels of autonomous MMPs (Fig. 3d). Last, we measured the extent to which Ras^High/*Lkb1*^−/− cells invade local tissues by dissecting cephalic complexes and assaying extent of migration over the ventral nerve cord (VNC). Compared to Ras^High control tissue which exhibits benign overgrowths confined to the eye-antennal discs, Ras^High/*Lkb1*^−/− cells completely invade contiguous organs like the brain hemispheres and VNC (Fig. 3g). These data suggest Ras^High/*Lkb1*^−/− tumor cells escape the basement membrane using an active proteolytic process and invade local tissues.

Invasion and metastasis are difficult processes to visualize in living organisms. Thus far, *Drosophila* tumor-bearing larvae have been precluded from fast, high resolution long-term intravital imaging techniques due to their size, degree of movement, and light scattering throughout the body due to the larval cuticle. To address this, we prepared live tumor-bearing larvae for long-term

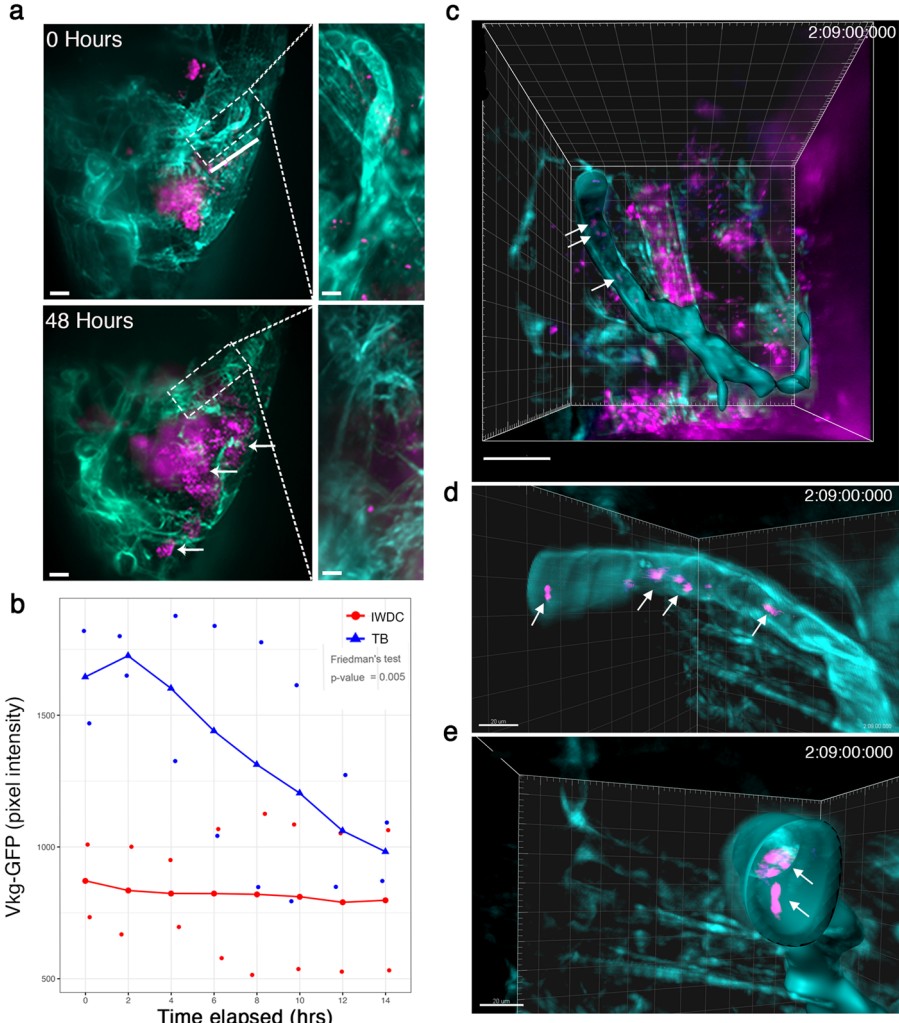

**Fig. 4 SiMView light-sheet microscopy allows visualization of collagen IV degradation by tumor cells over time. a** Maximum intensity projection from a 48 h SiMView imaging session on the anterior end of a $Ras^{High}/Lkb1^{-/-}$ tumor-bearing 'giant' larva (13 days AEL). Mutant cells express RFP (magenta) and Vkg-GFP (collagen IV-GFP, teal) is expressed throughout the organism. Scale bar, 20 μm. White dashed box is a representative region of interest (ROI, tracheal branch) and is magnified in the panels on the right. Scale bar, 300 μm. White arrows indicate RFP-positive $Ras^{High}/Lkb1^{-/-}$ cells that have invaded dorsally. **b** The effect of tracheal branch (TB; blue) vs. internal wing disc control (IWDC; red) groups on Viking-GFP pixel intensity on time elapsed. The group effect was measured using Friedman's test. **c** An Imaris Surface object of *Vkg-GFP* (teal) was generated from the ROI (above) using min and max thresholds of 250 and 385, respectively. White arrows indicate RFP-positive tumor cells (magenta) that appear embedded within the tracheal collagen matrix. **d, e** Zoom and rotated data channels were duplicated with voxels outside the Imaris object set to 0 in order to allow for better visualization with a maximum intensity projection view and clipping plane to show presence of RFP-positive cells within the tracheal matrix. Scale bar, 20 μm.

intravital imaging and used simultaneous multiview (SiMView) light-sheet microscopy[33] to visualize tumor cell and collagen IV dynamics for up to 48 h. SiMView allowed imaging of rapid cellular processes over time on an organismal scale, with minimal photobleaching. We collected image stacks in the *z* range every 60-s on two individual 'giant' tumor-bearing larvae with RFP-tagged $Ras^{High}/Lkb1^{-/-}$ mutant cells and Vkg-GFP expressed in the basement membrane of all epithelial tissues. Breakdown of Vkg-GFP was visible over time in each individual larva, especially in overlying tracheal branches dorsal to the tumor surface (Fig. 4a and Supplementary Movie 1). We defined two independent regions of interest in each larva that encompassed a tumor-adjacent tracheal branch and calculated Vkg-GFP pixel intensity every 2 h over a 14 h imaging window. Using the wing disc of each animal as an internal control, we observed a statistically significant difference in the change in levels of Vkg-GFP over the imaging window in the tracheal branches (Fig. 4b). Volumetric rendering and surface reconstruction of the tracheal branches

revealed tumor cells in contact with trachea at several hundred μm away from the primary tumors (Fig. 4c–e) and on rare occasions were found on the 'interior' surface of Vkg-GFP. These data suggest $Ras^{High}/Lkb1^{-/-}$ mutant cells actively invade tracheal vascular cells to potentially spread to distant organs.

**$Ras^{High}/Lkb1^{-/-}$ malignant tumors depend on CaMK/Ampk signaling in vivo.** Targeting effector signaling in KRAS-driven non-small cell lung cancer has resulted in limited efficacy in the clinic. In addition, previous studies have highlighted the additional complex transcriptional and signaling network changes in KRAS-driven tumors co-mutated for the tumor suppressor *LKB1*[5]. Therefore, rapid and genetically tractable models of Kras/Lkb1 tumors may shed light on the complex rewiring of signaling pathways and highlight novel targeting approaches. To probe effector pathways in our tumor model we used Western analysis on a panel of *Drosophila* epithelia harboring mutant clones for $Ras^{Low}$, $Ras^{Low}/Lkb1^{-/-}$, $Ras^{High}$, and $Ras^{High}/Lkb1^{-/-}$. Similar

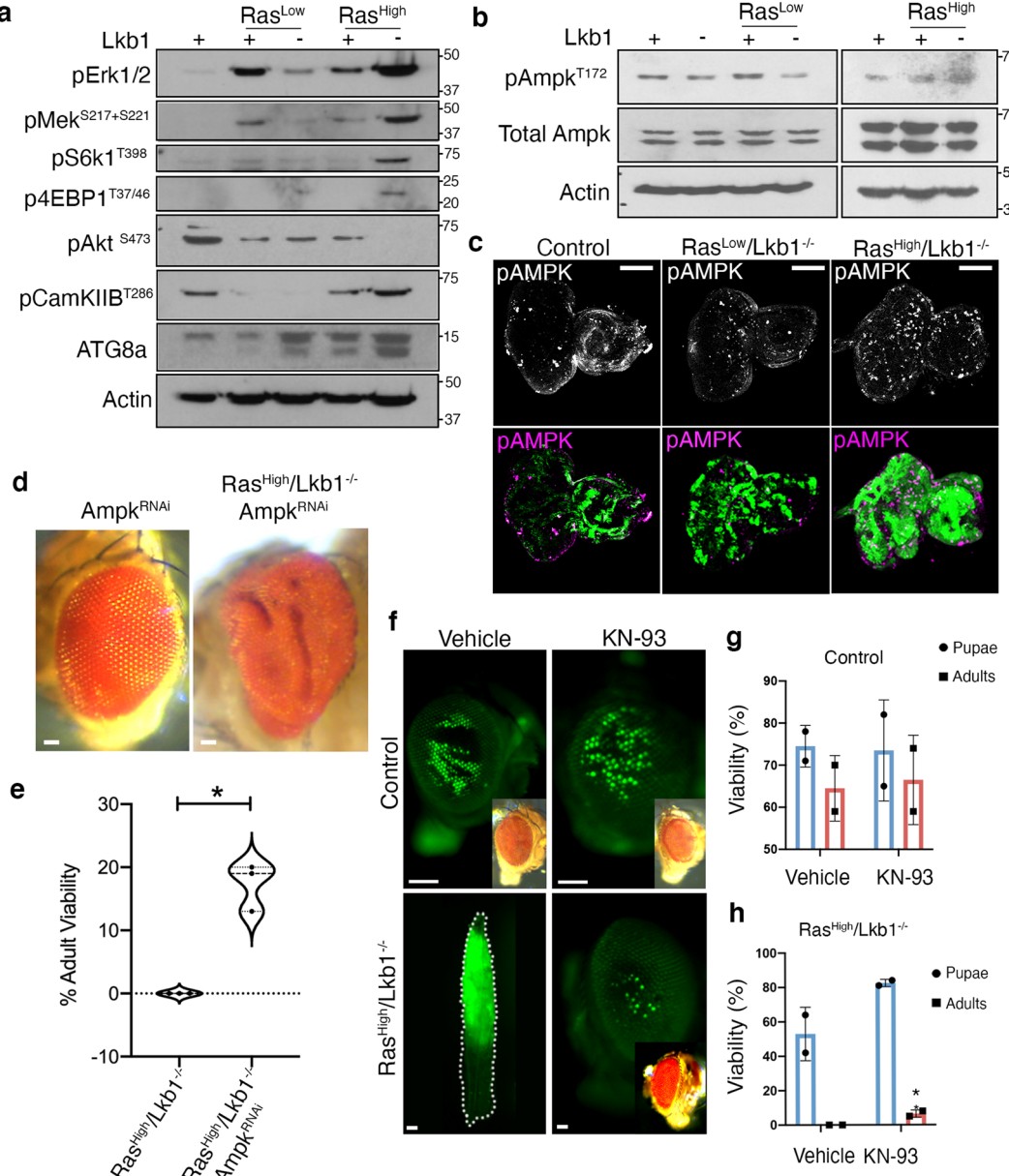

**Fig. 5 Neoplastic Ras^High^/*Lkb1*^−/−^ tumors depend on the genetic dose of ampk and are targetable with a CaMK inhibitor. a** Western analysis to assay activation of the indicated molecular pathways in mosaic larval eye-imaginal discs of the indicated genotypes. **b** Western analysis of Ampk activation from mosaic larval eye-imaginal discs of the indicated genotypes. **c** Confocal images of eye-imaginal discs carrying GFP+ clones of the indicated genotypes (control = FRT82B), and stained for phosphorylated Ampk. Images are representative of $n = 10$ independent eye-imaginal discs per genotype. Scale bar, 100 μm. **d** Representative brightfield images of adult eyes carrying clones of the indicated genotypes. **e** Violin plot showing % adult viability from the Ras^High^/*Lkb1*^−/−^ ($n = 204$) and Ras^High^/*Lkb1*^−/−^/*ampk*^RNAi^ ($n = 139$) genotypes from three independent experiments, *p-value = 0.0155 using a Welch's t-test. Dashed line represents the median, dotted lines represent the upper and lower quartiles. **f** Representative fluorescent and brightfield (inset) images of either flies carrying control (FRT82B) or GFP+ Ras^High^/*Lkb1*^−/−^ clones that were pharmacologically treated with vehicle or the pan CaMK inhibitor KN-93 (5 μM) as 1st instar larvae. Scale bars, 100 μm. **g, h** The percent survival to pupal and adult stages was quantified for control (FRT82B) and Ras^High^/*Lkb1*^−/−^ vehicle and KN-93 treated larvae. Data are represented as mean percent survival with error bars representing standard deviation. $n = 50$ biologically independent animals/genotype/condition/2 independent experiments. *p-value = 0.0493 computed from Unpaired t-test.

to human *KRAS/LKB1* tumors, increases in the activation of the RAS effector circuit Erk/Mek were observed along with S6K and 4EBP1 suggesting increased mTOR pathway activity (Fig. 5a and Supplementary Fig. 5). Compared to all other genotypes AKT is not active in Ras^High^/*Lkb1*^−/−^ cells most likely owing to sustained pS6K signaling resulting in a negative feedback loop by ribosomal protein S6. Previous studies have attributed increased TOR pathway activity in *LKB1* mutant tissue to loss of mTOR pathway inhibition by AMPK[34]. Therefore, we tested for loss of AMPK

activity in our panel of *Lkb1*^−/−^ mutant *Drosophila* tissue. We observed basal activation of Ampk in control tissue, followed by minimal activation in Ras^Low^/*Lkb1*^−/−^ mutants, most likely resulting from the overgrowth of surrounding wild-type epithelial tissue (Fig. 5b and Supplementary Fig. 5). However, in Ras^High^/*Lkb1*^−/−^ tissue we observed sustained pAmpk levels by Western blot, which was confirmed by immunofluorescence in mosaic imaginal discs (Fig. 5b, c). Recently, the presumed role of Ampk as a tumor suppressor has been challenged by evidence that

Ampk can promote metabolic adaptation to effect tumor growth and survival[35]. To test whether Ras$^{High}$/Lkb1$^{-/-}$ tumors are dependent on the genetic dose of *ampk* we expressed an RNAi transgene to *ampk* (knockdown efficiency of 50%; Supplementary Figs. 2a and 6) in developing GFP$^+$ Ras$^{High}$/Lkb1$^{-/-}$ tissue. Inhibition of Ampk via RNAi in Ras$^{High}$/Lkb1$^{-/-}$ mutant clones resulted in a statistically significant percentage of flies surviving to adulthood (Δ mean = +8; 95% CI, 3.697–12.3) (Fig. 5e). Interestingly, surviving flies exhibited hyperplastic overgrowth similar to that of Ras$^{Low}$/Lkb1$^{-/-}$ adult flies (compare Figs. 5d to 1c). A recent study from the Guo group found that autophagy may sustain AMPK activity upon *Lkb1* loss to support tumor growth[36]. In support of this, we detected increased lipidated ATG8a in Ras$^{High}$/Lkb1$^{-/-}$ tumors, indicative of an increase in autophagic flux (Fig. 5a and Supplementary Fig. 5). Altogether, these data support the conclusion that activation of Ampk is maintained in Ras$^{High}$/Lkb1$^{-/-}$ tumors and is required autonomously to promote malignant progression of *Kras/Lkb1* tumors in vivo.

The Ca$^{2+}$/calmodulin-dependent protein kinase kinase (CaMKK2) is a nucleotide-independent activator of AMPK[37], therefore we assayed activation of the *Drosophila* ortholog CamkIIB (48% identical/63% similar to CaMKK2) in our panel of mutant tissue. We found that activation of CamkIIB was elevated in Ras$^{High}$/Lkb1$^{-/-}$ tumors (Fig. 5a and Supplementary Fig. 5), suggesting a conserved role for this kinase in activating Ampk in the presence of oncogenic Ras tumors lacking Lkb1. To test whether Ras$^{High}$/Lkb1$^{-/-}$ tumors are dependent on CamkIIB activity we used pharmacologic inhibition of the CaMK cascade by feeding developing Ras$^{High}$/Lkb1$^{-/-}$ larvae with the inhibitor KN-93[38], which in our model inhibited activation of the *Drosophila* CamkIIB by 47% (Supplementary Figs. 2b and 6). Treatment of Ras$^{High}$/Lkb1$^{-/-}$ larvae resulted in a rescue of whole-organismal lethality, with an increase in the number of flies surviving to the pupal and adult stage (6.5% adult survival for KN-93 vs. 0% adult survival for vehicle control) (Fig. 5f–h and Supplementary Data 4). Taken together, these data suggest that in the context of loss of Lkb1, high levels of oncogenic Ras result in activation of Ampk by the alternative sole *Drosophila* CAMKK2 ortholog. Moreover, our pharmacologic results suggest that targeting the upstream AMPK/CAMKK complex may offer therapeutic benefit to *KRAS/LKB1* mutant lung adenocarcinoma patients.

**High levels of oncogenic KRAS and loss of *LKB1* result in decreased patient survival and AMPK signaling circuit activation in the TCGA lung adenocarcinoma cohort.** To test the translational relevance of our findings in *Drosophila* we analyzed human lung adenocarcinoma genomic and clinical data using cBioPortal[39,40] to study how differences in levels of oncogenic KRAS affect tumor progression in *LKB1* mutant patients. We used the TCGA Lung Adenocarcinoma PanCancer Atlas and TCGA Provisional Lung Adenocarcinoma datasets to select the proportion of patients with *KRAS* mutations in codon 12 (G12C, G12D, or G12V) for further study. We then used available RNA-sequencing data to stratify patients as either KRAS$^{Low}$ or KRAS$^{High}$. We next investigated overall patient survival by comparing cohorts of KRAS$^{Low}$ or KRAS$^{High}$ alone, to those that contained mono, biallelic loss, and/or loss-of-function mutations in *LKB1*. We found no difference in overall survival in KRAS$^{Low}$/LKB1$^{Mut}$ vs. KRAS$^{Low}$ patients (HR 2.181 95% CI, 0.9136–5.205), but strikingly KRAS$^{High}$/LKB1$^{Mut}$ patients exhibited significantly worse overall survival when compared with RAS$^{High}$ patients (HR 2.72; 95% CI, 1.132–6.546) (Fig. 6a, b). We then tested whether *KRAS* copy number changes could account for the change in

overall survival. Similar results were obtained when patients were stratified into either oncogenic KRAS$^{Diploid}$ (HR 2.048; 95% CI, 0.8241–5.090) or KRAS$^{Gain/Amp}$ (HR 4.993; 95% CI, 2.057–2.12) (Fig. 6c, d). Interestingly, the ability of high level vs. low-level KRAS to drive survival differences did not extend to patients with *TP53* mutations (Supplementary Fig. 3).

A recent study has reported that AMPK has a pro-tumorigenic role in lung cancer genetically engineered mice with Kras and p53 mutations[41]. Moreover, data from our *Drosophila Lkb1* mutant tumor model indicate that halving the genetic dose of *ampk* is sufficient to partially reverse whole-organism lethality. To test whether AMPK signaling may be involved in human *KRAS/LKB1* mutant lung adenocarcinoma we performed a correlation analysis between pAMPK and oncogenic codon 12 KRAS mRNA for *LKB1* loss-of-function and *LKB1* wild-type patients using TCGA data. We detected a positive correlation trend between pAMPK and oncogenic KRAS levels, but only in *LKB1* mutant patients (Spearman's correlation coefficient = 0.3, *p* = 0.068 for LKB1 loss-of-function vs. coefficient = −0.076, *p* = 0.683 for LKB1 wild-type patients) (Fig. 6e, f). To further test our hypothesis, we used canonical circuit activity analysis[42] which recodes gene expression data into measurements of changes in the activity of signaling circuits, ultimately providing high-throughput estimations of cell function. We performed the analysis to estimate the activity of the AMPK pathway in KRAS$^{High}$/LKB1$^{Mut}$ lung adenocarcinoma patients compared to KRAS$^{High}$ patients. The activity of three effector circuits is significantly (FDR < 0.05) upregulated in KRAS$^{High}$/LKB1$^{Mut}$ patients, one ending in the node that contains PPARGC1A (encodes PGC1alpha), the second one ending in the node with the MLYCD gene, and the third ending in the node containing EIF4EBP1 (Fig. 6g). These three genes control the cellular processes of circadian control of mitochondrial biogenesis, fatty acid metabolism, and translation regulation, and are known to be upregulated in various cancers[43–45]. These data confirm the translational relevance of our *Drosophila* model, and suggest that high oncogenic *KRAS* levels, perhaps through copy number gains and amplifications, activate specific sub-circuits of the AMPK signaling pathway to drive the malignant progression of *LKB1* mutant tumors.

## Discussion

It has been proposed that RAS-induced senescence functions as a tumor-suppressive mechanism[46]. More recent data have built upon these studies to show that high levels of Hras are required to activate tumor suppressor pathways in vivo[18], and that doubling the levels of oncogenic Kras is sufficient to cause metabolic rewiring leading to differences in therapeutic susceptibilities[19]. Mutant *Kras* copy gains are positively selected for during tumor progression in a *p53* mutant background[47]; however, our results analyzing survival in patients indicate that unlike *KRAS/LKB1* mutant patients, high levels of KRAS in *TP53*-mutant lung adenocarcinoma patients may not be a key factor in determining overall survival. In contrast, high-level KRAS and loss of *LKB1* leads to significantly decreased overall survival in lung cancer. Interestingly, *LKB1* has been shown to control genome integrity downstream of DNA damaging agents and cellular accumulation of ROS. Moreover, alterations in *LKB1* occur more frequently in patients with no known mitogenic driver[48]. Future work should uncover whether *KRAS* copy number gains and amplifications are positively selected for due to the role of LKB1 as a gatekeeper of genome integrity.

Seminal work in *Drosophila* identified the loss of epithelial polarity genes as key cooperating events in Ras-driven tumors in vivo[13,14]. In addition to its role in regulating cell growth, the Lkb1 protein is required to establish and maintain cell polarity

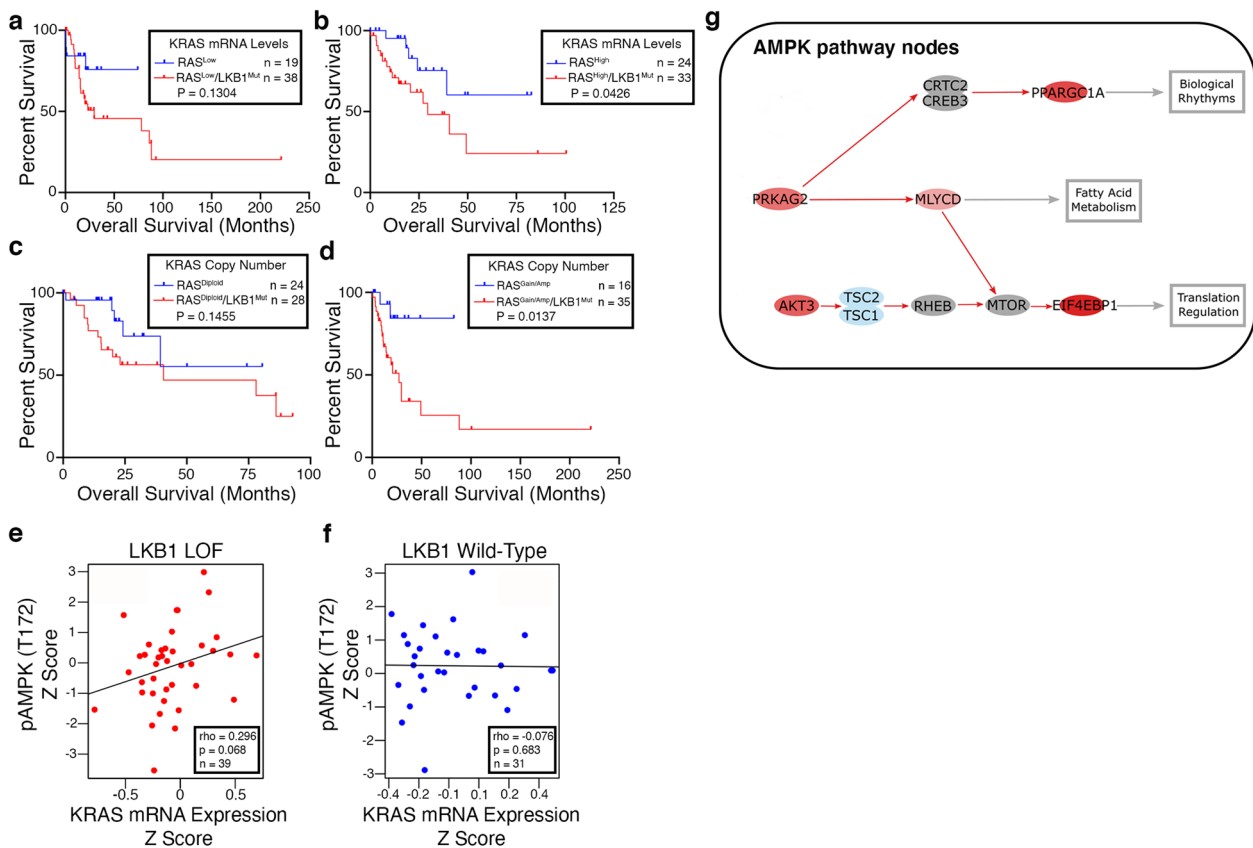

**Fig. 6 High levels of oncogenic KRAS drive decreased patient survival and is associated with AMPK activation in *LKB1* mutant patients. a**, **b** Analysis of patient survival using the TCGA Pan Lung Cancer study. Kaplan–Meier plots stratified by KRAS^Low (**a**) or KRAS^High (**b**) using oncogenic (codon 12) KRAS mRNA expression and further stratified based on *LKB1* deletion and loss-of-function mutation status. **c**, **d** Analysis of patient survival using the TCGA Pan Lung Cancer study. Kaplan–Meier plots stratified by KRAS^Low or KRAS^High using oncogenic (codon 12) *KRAS* copy number data and further stratified based on *LKB1* deletion and loss-of-function mutation status. **e**, **f** Analysis of phosphorylated AMPK (T172) expression as it correlates with KRAS mRNA expression and *LKB1* mutation status. **g** Canonical circuit activity analysis was used to estimate the activity of AMPK signaling pathway (hsa04152) that result in functional cell activities. Red color represents significantly ($p < 0.05$) upregulated genes (or paths) in KRAS^High/*LKB1*^Mut lung adenocarcinoma patients with respect to KRAS^High patients, and blue represents downregulated genes (or paths). The activity of three effector circuits is significantly (FDR $< 0.05$) upregulated in KRAS^High/*LKB1*^Mut patients, one ending in the node that contains the protein PPARGC1A ($p = 0.005$; FDR $= 0.037$; Uniprot function Biological rhythms/Mitochondrial biogenesis), the second one ending in the node with the MLYCD protein ($p = 0.0064$; FDR $= 0.045$; Uniprot function Fatty acid metabolism), and the third ending in the node containing EIF4EBP1 ($p = 0.001$; FDR $= 0.013$; Uniprot function Translation regulation).

across eukaryotes. However, alleles of *Lkb1* were not reported to synergize with oncogenic Ras in these studies, the reason possibly due to insufficient oncogenic Ras levels. The fact that loss of *Lkb1* behaves differently than other known polarity mutants suggests that an alternate function underlies the aggressive nature of *Lkb1* mutant cancer. Our work for the first time shows that functional Ampk activity is required for the malignant progression of Ras^High/*Lkb1*^−/− tumors in vivo. Moreover, our findings in lung adenocarcinoma patients suggest that increased oncogenic KRAS is associated with increased activation of pAMPK in *LKB1* mutant patients. The fact that pharmacologic inhibition of the *Drosophila* Camkk2 ortholog using the compound KN-93 resulted in partial suppression of Kras^High/*Lkb1*^−/− larval/pupal lethality further suggests that high-level oncogenic signaling engages the CaMK pathway to activate Ampk in Ras/*Lkb1* mutant tissue.

Using canonical circuit activity analysis, we discovered PPARGC1A, which encodes the protein PGC1α, as significantly upregulated in KRAS^High/*LKB1*^Mut lung adenocarcinoma patients. Interestingly, studies in human prostate cancer have discovered metabolic adaptations through PGC1α-mediated mitochondrial biogenesis in response to CAMKKβ/AMPK signaling[49,50]. Future studies should focus on whether similar

adaptations drive tumor growth and survival in *KRAS/LKB1* mutant lung adenocarcinoma. In addition, work is needed to elucidate the mechanism used by high-level Ras signaling to engage the CaMK pathway. Last, our work is the first to show that Ampk can have a pro-tumorigenic role in *Lkb1* mutant cancer in vivo, and suggests that *KRAS/LKB1* mutant lung adenocarcinoma patients may benefit from CAMKK inhibitors.

## Methods

***Drosophila* stocks and maintenance.** Flies were grown on a molasses-based food at 25 °C.

The following *Drosophila* stocks were used: (i) $w^{1118}$; *FRT82B*, (ii) *Df(3R) Exel6169,P{XP-U}Exel6169/TM6B,Tb* (#7648), (iii) *UAS-RasV12, FRT82B* (RAS^High —modified from stock #4847), (iv) *FRT82B, UAS-P35*, (v) *UAS-Ampk^Trip20(RNAi)* (#57785)—all provided by the Bloomington *Drosophila* Stock Center. *UAS-RasV12; FRT82B* (RAS^Low)[14] was provided by Tian Xu. $w^{1118}$ and *Viking-GFP*[51] were gifts from K. Moberg (Emory University). *Lkb1^4A4-2* and *Lkb1^4B1-11* were gifts from J. McDonald (Kansas State University). *Lkb1^X5* was a gift from W. Du (University of Chicago). Fluorescently labeled mitotic clones were induced in larval eye-imaginal discs using the following strain: *y,w, eyFLP1; Act >y+ >Gal4, UAS-GFP (or RFP); FRT82B, Tub-Gal80* (provided by Tian Xu).

**Generation of *Drosophila* Lkb1 antibody.** ProteinTech was used to generate a custom Lkb1 polyclonal antibody specific to *Drosophila* using the following peptide sequence: VEDEMTVLLANKNFHYDV-Cys. Guinea Pigs were immunized and

supplemented with booster immunizations before final antibody production after 102 days. Antibodies were affinity purified with Elisa confirmation of purification, and final antibody concentrations were estimated by SDS-PAGE.

**BrdU staining**. Third instar larval eye-imaginal discs were dissected in Grace's Insect Medium (ThermoFisher) then transferred into Grace's Insect Medium containing 0.25 mg/ml BrdU (Invitrogen B23151) and incubated at 25 °C for 90 min. Discs were then washed in Grace's Insect Medium for 5 min on ice followed by washing two times for 5 min each in 1× PBS on ice. Discs were fixed overnight (wrapped in foil) in 1% paraformaldehyde/0.05% Tween20. The following day discs were washed three times for 5 min each in 1× PBS and permeabilized for 20 min at RT in 0.3% PBST. To remove detergent, discs were washed five times for 5 min each in 1× PBS and DNAse treated for 30 min at 37 °C. Discs were then washed three times for 10 min each in 0.1% PBST and incubated overnight at 4 °C in mouse anti-BrdU primary antibody (B44) (BD, 1:50). The next day, discs were washed five times for a total of 30 min with 0.1% PBST and incubated overnight in goat anti-mouse F(ab)'2 AlexaFluor-555 secondary antibody (Cell Signaling, 1:500). Finally, discs were washed three times for 10 min each in 0.1% PBST and mounted in VectaShield anti-fade mounting medium.

**Cell-cycle analysis**. Live GFP-labeled 3rd instar eye-imaginal disc cells were dissected in 1× PBS and simultaneously dissociated with gentle agitation and stained (wrapped in foil) with Hoechst 33342 (Cell Signaling, 500 g/ml) for 2 h using a solution of 450 μl 10× Trypsin-EDTA (Sigma), 50 μl 10× PBS, and 0.5 μl Hoechst 33342. Cells were then passed through a 40 μm cell strainer prior to FACS analysis. Hoechst 33342 expression was analyzed for a minimum of 10,000 GFP-positive cells by flow cytometry on a Becton Dickinson FACS Canto II cytometer using FACSDiva software. Elimination of dead cells and the distribution of cells within G1, S, and G2/M phases of the cell cycle was determined using FlowJo software.

**Western blotting**. Twenty 3rd instar larvae were dissected in 1× PBS and eye-imaginal discs were transferred to a 1.5 ml microcentrifuge tube containing 1 ml of fresh 1× PBS. Discs were spun down at 4 °C for 1 min at 9600 × g and supernatant was removed. Two times Laemmli Sample Buffer was added and discs were boiled for 10 min at 100 °C, and spun down. Approximately 10 μg of protein was loaded into a 12% polyacrylamide gel. Alternatively, 3rd instar larvae were dissected and 20 μg of crude extract was loaded into a 10% polyacrylamide gel. Samples were run at 100 V and separated by SDS-PAGE before transferring to a polyvinylidene difluoride (PVDF) membrane overnight at 0.07 amps at 4 °C. Membranes were blocked for 1 h with 10% skim milk in 1× tris-buffered saline plus Tween20 (TBST) and placed in primary antibody overnight in 1× TBST with 5% skim milk or BSA at 4 °C. The following day, membranes were washed three times for 10 min each in 1× TBST and placed in secondary antibody in 1× TBST with 5% skim milk or BSA for 1 h at RT. After three additional 10-min washes in 1× TBST, ECL-reagent (Amersham, RPN2232) and X-ray film were used to detect signals. When necessary, membranes were stripped using GM Biosciences OneMinute Plus Western Blot Stripping Buffer (GM6011). Primary antibodies and dilution: affinity purified guinea pig anti-*Drosophila* Lkb1 (ProteinTech, 1:1000), rabbit anti-Ras (Cell Signaling 3965, 1:1000), rabbit anti-phospho Ampk (Thr 172) (40HP) (Cell Signaling, 1:1000), mouse anti-*Drosophila* AMPK1/2 (BioRad, 1:1000), rabbit anti-diphosphorylated ERK (Sigma, 1:1000), rabbit anti-phospho MEK1 (Ser 217 + 221) (Invitrogen, 1:500), rabbit anti-*Drosophila* phospho p70 S6 Kinase (Thr 398) (Cell Signaling, 1:1000), rabbit anti-phospho 4EBP1 (Thr 37/46) (Cell Signaling, 1:1000), rabbit anti-phospho AKT (Ser 473) (Cell Signaling, 1:1000), mouse anti-phospho CaMKII (Thr 286) (22B1 Santa Cruz Biotechnology, 1:200), rabbit anti-ATG8a (Creative Diagnostics, 0.2 g/ml), and mouse anti-actin (JLA20) (Developmental Studies Hybridoma Bank, 1:1000).

**Immunostaining**. Third instar larval eye-imaginal discs were dissected in 1× phosphate-buffered saline (PBS) and fixed in 4% paraformaldehyde for 30 min on ice. Discs were then washed three times for 10 min each in ice-cold 1× PBS, permeabilized in 0.3% Triton X100/1× PBS (PBST) for 20 min at RT, and washed again three times for 10 min each before blocking in 10% normal goat serum in 0.1% PBST for 30 min at RT. Discs were incubated in primary antibodies (4 °C overnight) in 10% normal goat serum/0.1% PBST. The following day, discs were washed three times for 5 min each in 0.1% PBST before incubating in secondary antibodies (in the dark at RT for 1 h) in 10% NGS/0.1% PBST. Finally, discs were washed three times for 10 min each in 1× PBS at RT and mounted using Vecta-Shield anti-fade mounting medium. Primary antibodies and dilution: rabbit anti-cleaved *Drosophila* DCP1 (Asp216) (Cell Signaling, 1:100), mouse anti-MMP1 (3A6B4/5H7B11/3B8D12 antibodies were mixed in equal amounts) (DSHB, 0.2 μg/ml), and Rabbit anti-pAMPK (T172) (Cell Signaling 1:100). Fluorescent secondary antibodies were from Life Technologies. DAPI was used to stain DNA.

**Widefield and confocal imaging**. Brightfield adult images were taken using a Leica S6D dissecting microscope. Fluorescent images were taken on a Leica MZ10F (×1 0.08899 NA) or Leica TCS SP8 inverted confocal microscope (×10 air HC PL Fluotar, 0.3 NA, ×20 air HC PL APO, 0.75 NA, or ×40 oil HC PL APO, 1.30 NA) using 0.88 μm z-stack intervals and sequential scanning (405 nm DMOD Flexible,

488 nm argon, 514 nm argon). All images were processed using ImageJ/FIJI and compiled in Adobe Photoshop.

**Allografting**. Tissue allografting was performed as described previously[52]. Third instar larvae were placed in a sterile petri dish containing 1× PBS and washed to remove residual fly food from the larval cuticle. Larval eye-imaginal discs were then dissected in 1× PBS. Sterile forceps were used to mince tissue into small pieces in preparation for implantation. $w^{1118}$ virgin female host flies were anesthetized with $CO_2$ and placed ventral-side up on double-sided sticky tape. Care was used to ensure that flies were well adhered to tape. A 10 μl sterile Hamilton Syringe with a 34 gauge 1-inch needle (45° needle angle) was used to aspirate a single piece of eye disc tissue into the needle, loading as little 1× PBS as possible. Forceps were used to hold the host abdomen steady and the syringe needle was used to pierce the abdomen and inject the eye disc tissue. Host flies were then removed from the double-sided tape and moved to a fresh vial of food placed horizontally at all times. Between genotypes the needle was cleaned by pipetting in and out with 1× PBS several times. Flies were monitored daily for survival and GFP-positivity, with transfer to new vials every two days. Death observed during the first 7 days was deemed artefactual, due likely to the injection procedure and not malignant growth. Flies were monitored for a total of 32 days.

**SiMView light-sheet microscopy**. Prior to mounting, live wandering 3rd instar *Drosophila* larvae and giant larvae (13 days AEL) were selected for stage and proper expression then cooled in a petri dish placed on top of an ice bucket. After sufficiently cooled to minimize movement, the samples were attached posterior side up to a 3 mm diameter stainless steel post using gel-control super glue (Ultra Gel Control, Loctite). When mounting, the sample's mouthparts were adhered in an extended state in order to improve image quality (i.e. reduce object depth) of the tumors. After allowing the adhesive to dry, the sample and post were loaded into an adapter that is magnetically attached to a multi-stage stack with degrees of freedom in the X–Y–Z and rotational directions. The sample chamber is sealed using custom-made rubber gaskets and filled with Schneider's Medium. The instrument is constructed as previously published with slight modification[33,53]. All data were collected using a Nikon 16×/0.8 NA LWD Plan Fluorite water-dipping objective and Hamamatsu Orca Flash 4.0 v2 sCMOS cameras. Exposure time for all experiments was 15 ms per frame. We collected data using a single camera view and two illumination arms, exciting with each arm in sequence for each color and timepoint. In our SiMView implementation for one-photon excitation, multiview image stacks are acquired by quickly moving the specimen over the desired z range and alternating light-sheet activation in the two illumination arms for each volume. This bidirectional illumination and detection capture recordings from two complementary views of each z plane in two illumination steps. Notably, no mechanical rotation of the specimen is required. The switching of laser shutters in the two illumination subsystems is performed within a few milliseconds. GFP and RFP fluorophores were excited using 488 and 561 nm Omicron Sole lasers, respectively.

**Analysis of SiMView data**. Following data acquisition, images were processed prior to analysis. All data had 90 counts subtracted to account for dark counts of the sCMOS cameras. Images from each illumination arm corresponding to the same Z slice were merged and corrected for intensity variation. Details on these algorithms are previously published[54]. Vkg-GFP pixel intensity over time was measured by using maximum intensity projections of 3D volumes from eight different time points between 0 and 14 h. The pixel intensity for the tracheal region of interest was measured for each timepoint using FIJI/ImageJ. 3D volumetric time-lapse data were visualized using Bitplane Imaris 9 (Fig. 4c–e). Subsets of the entire 2000–3000 timepoint series (~3–5 TBs in size) were selected for 3D inspection and visualization from maximum intensity projection images. 3D regions of interest (3D-ROI) were created using Imaris' intensity-based Surfaces function.

**Pharmacology**. Molasses-based food was melted and 10 ml of food was aliquoted to vials. While warm, 10 μl of $H_2O$ or 10 μl of 5 mM KN-93 (Millipore Sigma, 422711) were added to vials, respectively. Food vials were cooled and allowed to solidify before use. Vials not immediately used were placed at 4 °C. Adult *y,w, eyFLP1; Act > y + > Gal4, UAS-GFP; FRT82B, Tub-Gal80* virgin female flies were crossed to *FRT82B* or *UAS-RasV12High/Lkb1^{4A4-2}* males, respectively. Flies were moved to embryo 'egg-laying cups' and allowed to egg-lay onto grape juice agar plates at 25 °C. Flies were moved onto fresh agar plates every 24 h. After each 24 h period, embryos were collected using forceps and placed onto a fresh vial of food. Embryos were placed at 25 °C and allowed to hatch. Once of age, 2nd-instar larvae were collected and placed onto drug-containing media at 25 °C. Survival was quantified as the percentage of total embryos placed that survived to pupation and adulthood.

**Survival analysis of patient data**. cBioPortal was used to obtain survival, copy number, mRNA expression, and RPPA expression data available through the Cancer Genome Atlas (TCGA). For survival analysis, specific studies used included: TCGA Pan-Lung Cancer study[55] and TCGA Lung Adenocarcinoma studies (PanCancer Atlas[56] and Provisional). Out of 1144 total samples, samples with

specific KRAS G12C, G12D, or G12V mutations were selected for further analysis (115 samples for mRNA analysis and 76 samples for copy number analysis). Stratification as KRAS$^{Low}$ or KRAS$^{High}$ was based on normalized (Log2) mRNA expression or relative copy number. Patients with KRAS normalized mRNA expression value <10.825 were designated as RAS$^{Low}$, while patients with a normalized mRNA expression value >10.825 were designated as RAS$^{High}$. For copy number analysis, diploid patients were designated as KRAS$^{Diploid}$, while patients with KRAS gains and amplifications were designated as KRAS$^{Gain/Amp}$. Of these patients, mono or biallelic loss or predicted loss-of-function mutations in *LKB1* or deletions or loss-of-function mutations in *TP53* were also obtained. For pAMPK correlation analysis, specific studies used included: TCGA Pan-Lung Cancer study and TCGA Lung Adenocarcinoma study (PanCancer Atlas). Samples with specific KRAS G12C, G12D, or G12V mutations as well as RPPA expression data for pAMPK (T172) were selected for further analysis ($n = 71$).

**Canonical circuit activity analysis.** The HiPathia web-application (http://hipathia.babelomics.org) was used to identify differentially expressed (activated or inhibited) pathways. RNA-sequencing raw RSEM count data based on human genome build hg19 was obtained for the TCGA lung adenocarcinoma patients from the Genomic Data Commons (GDG) legacy archive (https://portal.gdc.cancer.gov/legacy-archive). Patients without all data types were excluded. Patients were obtained and grouped based on KRAS status using CBioPortal as previously described and determined to be KRAS/LKB1WT if concomitant somatic mutations in LKB1 were absent, and as KRAS/LKB1Mut if secondary LKB1 mutations were present. Genes with average counts per million of >0.1 across all samples were kept. Normalization was done with a trimmed mean of M-values method and log2 transformed using edgeR package[57]. The normalized expression matrix was then used as input in HiPathia web-application to identify up- or downregulated pathways between the two groups (mutant vs. wild type) against all available pathways in HiPathia. Finally, differential gene fold change was estimated using the Limma R package.

**Statistics and reproducibility.** Statistical tests, sample size, and number of biological replicates are reported in the figure legends. In summary, GraphPad Prism 7 and 8 were used to generate $P$ values using the two-tailed unpaired Student's $t$-test to analyze statistical significance between two conditions in an experiment, ordinary one-way ANOVA with a Tukey's multiple comparisons test for experiments with three or more comparisons, and Log-rank (Mantel-Cox) test for analysis of survival data. Significance was assigned to $p$ values <0.05 unless otherwise indicated. For Fig. 6e–g, statistical analysis was conducted using RStudio. For Fig. 6e and f, data were divided into two groups, LKB1 loss of function ($n = 40$) and LKB1 wild type ($n = 31$). A single outlier sample in the LKB1 mutation category was excluded and calculated $z$-score for pAMPK and KRAS expression data was used. The correlation between the AMPK and KRAS was conducted and a Spearman's correlation test. Due to the relatively small sample size, a $p$-value of ≤0.1 or 10% was considered significant.

**Reporting summary.** Further information on research design is available in the Nature Research Reporting Summary linked to this article.

## Data availability

The molecular and clinical data used to support the conclusions of Fig. 6 are available from the GDC data portal (http://portal.gdc.cancer.gov/). Source data for Fig. 1e, Fig. 2b, c, Fig. 3f, and Fig. 5g, h are available in Supplementary Data 1–4. All other data that support the findings of this paper are available from the corresponding author upon request and are stored locally in a research-grade peta-byte storage account through the Emory Integrated Computational Core.

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

## Acknowledgements

The authors thank T. Xu, J. McDonald, K. Moberg, and D. Lerit for gifts of fly stocks, reagents, and equipment. We acknowledge the Bloomington Drosophila Stock Center, Vienna Drosophila Resource Center, TRiP at Harvard Medical School (NIH/NIGMS R01-GM084947), and the Developmental Studies Hybridoma Bank (DSHB) for providing fly stocks and antibodies. We thank members of the A. Marcus, W. Zhou, K. Moberg, and R. Read laboratories for helpful comments, discussion, and teaching of techniques. This work was supported in part by the Advanced Imaging Center at the Janelia Research Campus of the Howard Hughes Medical Institute. The Advanced Imaging Center is jointly supported by the Gordon and Betty Moore Foundation and Howard Hughes Medical Institute. Research reported in this publication was supported in part by the Winship Biostatistics and Bioinformatics and Emory Integrated Cellular Imaging Shared Resources of Emory University under NIH/NCI award number P30CA138292. Research reported in this publication was supported by the National Cancer Institute of the National Institutes of Health under Award Number P50CA217691, R01CA194027 (MGR), and R01CA201340 (MGR). The content is solely the responsibility of the authors and does not necessarily represent the official views of the National Institutes of Health.

## Author contributions

M.G.R., B.R., C.S., and E.K. conceived and designed the project. B.R and C.S. performed the *Drosophila* and molecular biology experiments. R.E.P. contributed to visualization and editing of the manuscript. E.K., N.A., J.M.H., T.L.C., and M.G.R. designed and performed the SiMView imaging experiments. W.G., N.A., E.K., and M.G.R. analyzed the SiMView data. B.D., M.R., and B.R. performed the bioinformatic, correlation studies, and statistical analysis using human patient data, and B.R. and M.G.R. wrote the manuscript.

## Competing interests

The authors declare no competing interests.
