## [Peer Review File · Communications Biology]

Reviewers' comments:

Reviewer #1 (Remarks to the Author):

Rackley et al, present a novel *Drosophila* co-operative tumourigenesis model generated by over expression of oncogenic Ras and loss of the cell polarity tumour suppressor Lkb1. It is not surprising that expression of oncogenic Kras and loss of the tumour suppressor Lkb1 results in co-operative tumourigenesis in *Drosophila* as is acknowledged by the authors this has been observed in a mouse lung cancer model previously.

However the beauty of the *Drosophila* genetic model is the tractability of the tumours and they utilise simultaneous Multiview light-sheet microscopy of Kras^{Hi} Lkb1^{-/-} tumours to visualise cell migration in real time in vivo. They also perform molecular analysis to dissect the downstream consequences on different signalling pathways and their potential contributions to the tumour progression. With their conclusions being Kras^{Hi} Lkb1^{-/-} tumours activate mTOR and AMPK pathways with a dependence on CaMK activating AMPK.

The paper utilises two different models with high and low levels of oncogenic Kras and Lkb1 loss which result in different tumour phenotypes, to highlight the relevance of their model to human disease they stratify human lung cancer patients with high or low KRAS and loss of LKB1 and see a similar effect on disease severity. Specifically for tumours mutant for LKB1 Ras expression and pAMPK activity was also positively correlated supporting their conclusion that AMPK pathway plays a role in these patients and may provide a new therapeutic target.

Major point to address:

1. The development of long-term intravital imaging using SiMView light-sheet microscopy provides a new insight into in vivo tracking of tumour cells in the live *Drosophila* larvae which may provide the opportunity for great insight into the process of metastasis. My questions really revolve around how many observations were performed to generate the data presented? It appears n=1 for the data in Figure 3 which precludes any real conclusion on the rate of collagen degradation and this will depend on the age of the larvae/size of the tumour clones at the start of imaging. I appreciate the wing disc is used as an internal control region but I feel a much better control would be a clonal system which also results in giant larvae but in tumours that fail to invade. Many studies in the field have examples of this in terms of whole mount IF dissecting eye discs with the ventral nerve cord. Examples would be co-expressing BSKDN to block JNK signalling or combining oncogenic Ras with Tsc1^{-/-} (Igaki, *Current Biology*, 2006) and these would be potential controls for the Vkg-GFP. The same lack of number of observations and a control is true of the results presented in Figure 4 and the authors should discuss the limitation of their observations. I am not sure what this aspect of the study adds to the field in its current preliminary state.

Minor specific items to address to make paper clearer to reader and strengthen the conclusions made from the data:

1. The two UAS-RasV12 lines used and characterised as either Hi or Lo need to be appropriately referenced – only RasLo is referenced and RasHi is named as the same line in Materials and methods. Are they different strains? Catalogue numbers for Bloomington?

2. In line 135 the authors should state they assayed for CLEAVED death caspase 1 in mutant clones to specify activation of cell death pathway.

3. In Figure 2A the authors show examples of third instar eye imaginal discs and claim that the RasLo Lkb1^{-/-} tissue is larger but there is less GFP⁺ mutant clonal tissue. The represented examples in the figure do support this however in Figure 1 panel D the difference between the control and RasLo Lkb1^{-/-} does not seem very evident. Is this a good representative image?

4. In line 161 the authors state that altogether these data suggest 'they undergo G1 arrest while promoting the increased hyperplastic proliferation of surrounding WT tissue'. I believe this conclusion is correct and the FACS data in Supplemental Figure 2 is the evidence that supports it and so that data should be explained and referenced in this paragraph. Currently only the BrdU analysis is mentioned.

5. Where two examples are given of the representative clonal tissue of RasHi Lkb1^{-/-} tissue (Figure 2B), the authors should explain in the figure legend what 13d AEL means – I assume this is an example of the imaginal discs of the giant larvae at day 13 having failed to pupate.

6. The survival analysis following transplantation of tumour tissue in Figure 2G shows a significant survival disadvantage of RasLo Lkb1^{-/-} larvae as well but this is not discussed. The GFP levels do look a lot lower – is the tumour tissue still alive, growing, migrating? It should be acknowledged/discussed in light of the elevated MPP levels seen by IF in Figure 2E.

7. The pAmpk western in Figure 5C is not very convincing – if this has been done 3 times is this the best result achieved? The IF of pAMPK (5B) does support the conclusion but due to the unconvincing Western I think a RasHi Lkb1^{+/+} control is needed for the IF to claim this is specifically due to the absence of Lkb1. I think either stronger data needs to be presented or the wording in the text should honestly reflect what is shown.

8. Is the Ampk RNAi fly validated? If so reference, if not use the pAmpk antibody to show successful knockdown in clones by either western or IF.

9. Is KN-93 a well validated compound with on target effects or in clinical trials? Include references to support its use as a specific CAMKK inhibitor. If not again use western to validate on target effect on pathway after feeding larvae.

10. CCAA identified three sub circuits of the AMPK pathway upregulated in RasHi Lkb1^{-/-} compared to RasHi although the relevance of these to cancer was not clear. Are they targetable therapeutically?

I believe addressing the above comments will improve the paper but the major conclusions come from the molecular studies of Figure 5 and translating the finding to relevant human cancer patient cohorts. The conclusions of Figure 5 are that loss of Lkb1 in combination with high levels of oncogenic Ras drive co-activation of both mTOR and AMPK pathway however only AMPK pathway was followed up. The authors should extend their studies to include the contribution of mTOR in vivo or analyse it in terms of the human patient cohort data.

I think to conclude with the claim that the AMPK pathway is important and is activated through

CamK to induce autophagy for a pro-tumourigenic effect requires strengthening of the data in Figure 5 with further western blots, genetic or pharmacological interventions eg. Rapamycin to block mTORC1, metformin to block AMPK. Perhaps a schematic model of the pathway would help visualise how the authors believe the tumours are 're-wired' following loss of Lkb1 and high levels of oncogenic Ras to promote tumour growth.

The human lung patient data from the TCGA presented in Figure 6 supports their stratification of high and low KRAS levels and I realise both KRAS and LKB1 are highly mutated in lung cancer. I just wonder if the authors have looked for the association in any other type of cancer eg. Colon cancer - LKB1 important tumour suppressor and Pancreatic cancer - KRAS mutation major driver. This could further extend the interest of the study to researchers focused on other cancers and potentially open up new treatment targets for more patients.

Reviewer #2 (Remarks to the Author):

In this manuscript the authors address the heterogeneity and behavior of cells expressing the oncogenic form of ras (Kras) in combination with loss of Lkb1. In tumor cells this genetic combination may result in a metabolic heterogeneity of the ras cancer cells adding complexity to the therapeutic strategies applied in the clinic. The authors showed, using *Drosophila* that the combination of low level of expression of Kras promotes survival of Lkb mutant cells and this is accompanied to a non-autonomous growth of the surrounding tissues. Interestingly, when Kras was expressed at high levels Lkb mutant cells acquired a malignant phenotype and become transformed. This phenotype was analyzed for the ability of the malignant cells to be more invasive and to form small clones outside of the original clone environment. The activation of the TOR pathways with AMPK in the mosaic malignant Kras/Lkb1 tumors indicates a potential novel mechanism that correlates with the incidence of malignancy and the prognosis in human tumors.

The authors show the ability of using *Drosophila*'s genetics and imaging techniques to study genetically the interaction between components of signaling pathways (KRAS and LKB1) otherwise very difficult to analyze *in vivo* in vertebrates. I believe that the data and conclusions of this work help to better understand some phenotypes described in tumors characterized by similar mutations in KRAS LKB1 genes, and attempt to clarify some of the mechanisms that regulate the non-autonomous effect observed in these tumors. I therefore accept this paper for its publication in *Communications Biology*.

Minor changes:

Fig 2d: scale bar is misplaced

Fig 3d SD is missing

Fig 5a control loading is missing

Fig 5a the phospho-residues recognized by the antibodies should be written in the figure and the same is for 5c and d.

Fig 5e add the quantification of GFP and statistics

Fig 6 quality of image g must improve the words are not clearly visible at least from the printout submitted

Response to Referees for COMMSBIO-20-0113A

We would first like to thank both reviewers for their time and helpful comments. Our laboratory is located within the Winship Cancer Institute of Emory University, which houses outpatient cancer clinics and a chemotherapy infusion center. Due to COVID-19, access to my laboratory was prohibited shortly after receiving the initial reviews, and remained off-limits until June 1st. Since then, we have been able to operate with one person/250ft². Despite these setbacks, we have implemented creative solutions to fully address all comments. We believe our manuscript to be much improved. Below is a point-by-point summary of response to critiques:

Reviewer #1 (Remarks to the Author):

Major point to address:

1. The development of long-term intravital imaging using SiMView light-sheet microscopy provides a new insight into in vivo tracking of tumour cells in the live *Drosophila* larvae which may provide the opportunity for great insight into the process of metastasis. My questions really revolve around how many observations were performed to generate the data presented? It appears n=1 for the data in Figure 3 which precludes any real conclusion on the rate of collagen degradation and this will depend on the age of the larvae/size of the tumour clones at the start of imaging.

Response: We have added into our analysis an additional independent larva and imaging session. Both larvae were imaged at ~13d after egg lay (AEL) at 25-degrees and were 'giant larva'. The success of this technique depends on the GFP+ tumor clones having grown large enough to the dorsal cuticle – so each larva had GFP+ tumors that were approximately the same size. In addition, the rotation of each larva allowed us to simultaneously gather Vkg-GFP data from a wing disc, which we used as an internal control. Instead of making a conclusion about the rate, we have used a statistical test to compare the change in the amount of collagen over the imaging window for each group.

I appreciate the wing disc is used as an internal control region but I feel a much better control would be a clonal system which also results in giant larvae but in tumours that fail to invade. Many studies in the field have examples of this in terms of whole mount IF dissecting eye discs with the ventral nerve cord. Examples would be co-expressing BSKDN to block JNK signalling or combining oncogenic Ras with Tsc1^{-/-} (Igaki, Current Biology, 2006) and these would be potential controls for the Vkg-GFP.

Prior to our work, and as mentioned by Reviewer #1, active local invasion in other genotypes of neoplastic 'giant' Drosophila larvae has been inferred from the presence of breaks in GFP-tagged collagen IV using dissection, fixation, and standard widefield or confocal microscopy, or by assaying invasion into local tissues

such as the ventral nerve cord. For the purpose of our studies, and since we are interested in how Ras and Lkb1 cooperate in vivo to promote tumor progression, we believe the appropriate control to be larvae expressing RasHi alone in clones. Unfortunately, RasHi-expressing eye/imaginal discs (while much larger than wild-type eye discs) do not grow to the size required for successful imaging through the larval cuticle using SiMView. To address Reviewer #1's comments we have added standard assays in Figure 2 showing that Vkg-GFP is not compromised in RasHi discs (see Fig. 2c), nor is MMP elevated autonomously in clones (Fig. 2d). Lastly, we have added new data using the 'gold standard' assay in the field showing complete invasion of the VNC in RasHi/Lkb1 vs. no invasion for RasHi discs.

The same lack of number of observations and a control is true of the results presented in Figure 4 and the authors should discuss the limitation of their observations. I am not sure what this aspect of the study adds to the field in its current preliminary state.

Reviewer #1 is absolutely correct. We have removed the data and conclusions presented in the original Figure 4, which was not central to the main findings presented in the manuscript.

Minor specific items to address to make paper clearer to reader and strengthen the conclusions made from the data:

1. The two UAS-RasV12 lines used and characterized as either Hi or Lo need to be appropriately referenced – only RasLo is referenced and RasHi is named as the same line in Materials and methods. Are they different strains? Catalogue numbers for Bloomington?

We have appropriately referenced both RasHi and RasLo stocks in the Materials and Methods.

2. In line 135 the authors should state they assayed for CLEAVED death caspase 1 in mutant clones to specify activation of cell death pathway.

We have corrected this important oversight.

3. In Figure 2A the authors show examples of third instar eye imaginal discs and claim that the RasLo Lkb1^{-/-} tissue is larger but there is less GFP⁺ mutant clonal tissue. The represented examples in the figure do support this however in Figure 1 panel D the difference between the control and RasLo Lkb1^{-/-} does not seem very evident. Is this a good representative image?

Thank you for pointing this out. We have added confocal images of an appropriately aged disc that represents the hyperplastic phenotype of RasLo Lkb1^{-/-} tissue.

4. In line 161 the authors state that altogether these data suggest 'they undergo G1 arrest while promoting the increased hyperplastic proliferation of surrounding WT tissue'. I believe this conclusion is correct and the FACS data in Supplemental Figure 2 is the evidence that supports it and so that data should be explained and referenced in this paragraph. Currently only the BrdU analysis is mentioned.

Since the data in former Suppl. Fig. 2 is central to the findings of the manuscript, we have added the data to what is now Figure 2, and have explained and referenced the FACS data in the accompanying Results section.

5. Where two examples are given of the representative clonal tissue of Ras^{Hi} Lkb1^{-/-} tissue (Figure 2B), the authors should explain in the figure legend what 13d AEL means – I assume this is an example of the imaginal discs of the giant larvae at day 13 having failed to pupate.

Reviewer #1 is correct and the stage '13d AEL' has been appropriately explained in the figure legend.

6. The survival analysis following transplantation of tumour tissue in Figure 2G shows a significant survival disadvantage of Ras^{Lo} Lkb1^{-/-} larvae as well but this is not discussed. The GFP levels do look a lot lower – is the tumour tissue still alive, growing, migrating? It should be acknowledged/discussed in light of the elevated MPP levels seen by IF in Figure 2E.

We have addressed this in the text by postulating that we cannot rule out the evolution and potential partial transformation of the overgrown GFP- tissue from Ras^{Lo}/Lkb1^{-/-} discs.

7. The pAmpk western in Figure 5C is not very convincing – if this has been done 3 times is this the best result achieved? The IF of pAMPK (5B) does support the conclusion but due to the unconvincing Western I think a Ras^{Hi} Lkb1^{+/+} control is needed for the IF to claim this is specifically due to the absence of Lkb1. I think either stronger data needs to be presented or the wording in the text should honestly reflect what is shown.

Yes, despite repeating the pAmpk western multiple times using different conditions, this is the clearest band we can achieve. One aspect of the biology of these tumors is that they are full of acidic vesicles we believe to be autolysosomes. In fact, the tumors are so acidic that by day 18 AEL the GFP+ tumors have photoconverted and can be visualized using our RFP filters. Since Lkb1/Ampk signaling has been shown to occur on endosomes (O'Farrell et al. 2017), it is possible that the quality of the band is being affected by cell compartment and/or pH. We believe that the western, combined with the IF and genetic data together provide strong evidence that Ampk remains active in Ras^{Hi}/Lkb1^{-/-} tumors.

8. Is the Ampk RNAi fly validated? If so reference, if not use the pAmpk antibody to show successful knockdown in clones by either western or IF.

We have validated the percent knockdown of the ampk^{Trip20} line in our hands (see Suppl. Fig. 2)

9. Is KN-93 a well validated compound with on target effects or in clinical trials? Include references to support its use as a specific CAMKK inhibitor. If not again use western to validate on target effect on pathway after feeding larvae.

KN-93 is a well-validated inhibitor of multiple members of the CamK pathway. We have referenced an extensive review in the text to support its use. Since the sole Drosophila ortholog has conserved features of multiple CaMKs we thought this was the best strategy to capture 'proof-of-principle' data. Lastly, we have validated the compound in our hands using western analysis of tumors from KN-93-fed Ras^{H1}/Lkb1⁻ larvae (see Suppl. Fig. 2).

10. CCAA identified three sub circuits of the AMPK pathway upregulated in RasHi Lkb1^{-/-} compared to RasHi although the relevance of these to cancer was not clear. Are they targetable therapeutically?

We have updated the text to include references on the cancer-relevance of each upregulated circuit/node.

I believe addressing the above comments will improve the paper but the major conclusions come from the molecular studies of Figure 5 and translating the finding to relevant human cancer patient cohorts. The conclusions of Figure 5 are that loss of Lkb1 in combination with high levels of oncogenic Ras drive co-activation of both mTOR and AMPK pathway however only AMPK pathway was followed up. The authors should extend their studies to include the contribution of mTOR in vivo or analyse it in terms of the human patient cohort data.

LKB1 is a well-established negative regulator of mTOR signaling (reviewed in Momcilovic and Shackelford, 2015); however, single-agent rapalogs have shown no therapeutic benefit in Kras/Lkb1 lung cancer GEMMs, and have performed poorly in lung cancer clinical trials. For these reasons, we initially decided not to follow-up on the mTOR pathway. We have updated the conclusions of Figure 5 to focus solely on the CaMK/Ampk signaling pathway – which to our knowledge has yet to be pursued as a therapeutic vulnerability in Kras/Lkb1-mutant lung cancer.

I think to conclude with the claim that the AMPK pathway is important and is activated through CamK to induce autophagy for a pro-tumourigenic effect requires strengthening of the data in Figure 5 with further western blots, genetic or pharmacological interventions eg. Rapamycin to block mTORC1 (DONE), metformin

to block AMPK (This has already been published in mammalian models). Perhaps a schematic model of the pathway would help visualise how the authors believe the tumours are 're-wired' following loss of Lkb1 and high levels of oncogenic Ras to promote tumour growth.

Since rapamycin has not performed well in pre-clinical or clinical studies as monotherapy, and metformin is controversial due to its indirect effect on Ampk, we have removed the conclusion that $Kras^{Hi}/Lkb1^{-/-}$ tumors are 're-wired' metabolically. The lipidated Atg8 western blot remains in the manuscript to add further evidence that Ampk is active in these tumors.

The human lung patient data from the TCGA presented in Figure 6 supports their stratification of high and low KRAS levels and I realise both KRAS and LKB1 are highly mutated in lung cancer. I just wonder if the authors have looked for the association in any other type of cancer eg. Colon cancer - LKB1 important tumour suppressor and Pancreatic cancer - KRAS mutation major driver. This could further extend the interest of the study to researchers focused on other cancers and potentially open up new treatment targets for more patients.

We did investigate this. Since there is a paucity of co-mutated KRAS/LKB1-mutant tumors outside of lung cancer, we combined the data from pancreatic, colorectal, melanoma, uterine, and bladder cancer but saw no survival differences based on stratification of KRAS levels or copy number. The phenomenon is specific to lung cancer – and seems to be most pronounced for patients with KRAS codon 12 mutations (which of course are predominant in lung cancer).

Reviewer #2 (Remarks to the Author):

Minor changes:

Fig 2d: scale bar is misplaced

Fixed in what is now Figure 3c

Fig 3d SD is missing

We have included the S.E.M. in what is now Figure 4b

Fig 5a control loading is missing

The growth pathways and the CamKII and lipidated Atg8 western were all probed on the same membrane. They have been combined into one panel with the relevant Actin control blot.

Fig 5a the phospho-residues recognized by the antibodies should be written in the figure and the same is for 5c and d.

Done.

Fig 5e add the quantification of GFP and statistics.

We scaled up and independently repeated the ampk genetic rescue experiment three times. From our new expanded analysis, it became clear that the predominant phenotype of $Kras^{Hi}/Lkb1^{-/-}$ rescued adult flies was hyperplastic eyes with no GFP signal. For quantitation, we have calculated the percent adult viability (now Fig. 5d,e).

Fig 6 quality of image g must improve the words are not clearly visible at least from the printout submitted

We have improved the quality of the image in Fig. 6g to highlight the relevant signaling nodes.

REVIEWERS' COMMENTS:

Reviewer #1 (Remarks to the Author):

Rackley et al., present a *Drosophila* tumour model driven by oncogenic Ras and loss of Lkb1. The authors have clearly improved the manuscript since first submission and provided a detailed and complete response to all original suggestions and comments.

The major criticism of the reproducibility of the SiMView light-sheet microscopy has been addressed by repeating the analysis on an additional independent larvae. I think the paper nicely demonstrates the possible uses of this technology to visualise cell migration in real time in vivo and so will be of interest to many researchers.

Including the BrdU data and further controls into the new figure 2 (and supplemental figure 2) also help support and strengthen the conclusions and important finding of the paper highlighting the different outcome of hyperplastic growth dependent on High or Low Ras expression levels.

Furthermore, focussing on the novel finding in figure 5 that the CaMK/Ampk pathway is required for malignant RasHi Lkb1 tumours provides a strong message for the paper, the removal of the more speculative statements regarding 'metabolic re-wiring of the tumours' that I did not feel were strongly or clearly established simplifies the story. I think concluding with the observations from the TCGA analysis nicely supports the hypothesis that the findings of the paper may be relevant for human lung cancer patients with aggressive KrasHi Lkb1 mutant tumours. This new potential therapeutic vulnerability for this subset of cancers would be of interest to follow up.

I have no new criticisms of the data presented.

Reviewer #2 (Remarks to the Author):

I found this revised version of the manuscript improved and all my concerns have been addressed, therefore I accept the current version of the manuscript.

Response to Referees for COMMSBIO-20-0113A

We would first like to thank both reviewers for their time and helpful comments. Our laboratory is located within the Winship Cancer Institute of Emory University, which houses outpatient cancer clinics and a chemotherapy infusion center. Due to COVID-19, access to my laboratory was prohibited shortly after receiving the initial reviews, and remained off-limits until June 1st. Since then, we have been able to operate with one person/250ft². Despite these setbacks, we have implemented creative solutions to fully address all comments. We believe our manuscript to be much improved. Below is a point-by-point summary of response to critiques:

Reviewer #1 (Remarks to the Author):

Major point to address:

1. The development of long-term intravital imaging using SiMView light-sheet microscopy provides a new insight into in vivo tracking of tumour cells in the live *Drosophila* larvae which may provide the opportunity for great insight into the process of metastasis. My questions really revolve around how many observations were performed to generate the data presented? It appears n=1 for the data in Figure 3 which precludes any real conclusion on the rate of collagen degradation and this will depend on the age of the larvae/size of the tumour clones at the start of imaging.

Response: We have added into our analysis an additional independent larva and imaging session. Both larvae were imaged at ~13d after egg lay (AEL) at 25-degrees and were 'giant larva'. The success of this technique depends on the GFP+ tumor clones having grown large enough to the dorsal cuticle – so each larva had GFP+ tumors that were approximately the same size. In addition, the rotation of each larva allowed us to simultaneously gather Vkg-GFP data from a wing disc, which we used as an internal control. Instead of making a conclusion about the rate, we have used a statistical test to compare the change in the amount of collagen over the imaging window for each group.

I appreciate the wing disc is used as an internal control region but I feel a much better control would be a clonal system which also results in giant larvae but in tumours that fail to invade. Many studies in the field have examples of this in terms of whole mount IF dissecting eye discs with the ventral nerve cord. Examples would be co-expressing BSKDN to block JNK signalling or combining oncogenic Ras with Tsc1^{-/-} (Igaki, Current Biology, 2006) and these would be potential controls for the Vkg-GFP.

Prior to our work, and as mentioned by Reviewer #1, active local invasion in other genotypes of neoplastic 'giant' Drosophila larvae has been inferred from the presence of breaks in GFP-tagged collagen IV using dissection, fixation, and standard widefield or confocal microscopy, or by assaying invasion into local tissues

such as the ventral nerve cord. For the purpose of our studies, and since we are interested in how Ras and Lkb1 cooperate in vivo to promote tumor progression, we believe the appropriate control to be larvae expressing RasHi alone in clones. Unfortunately, RasHi-expressing eye/imaginal discs (while much larger than wild-type eye discs) do not grow to the size required for successful imaging through the larval cuticle using SiMView. To address Reviewer #1's comments we have added standard assays in Figure 2 showing that Vkg-GFP is not compromised in RasHi discs (see Fig. 2c), nor is MMP elevated autonomously in clones (Fig. 2d). Lastly, we have added new data using the 'gold standard' assay in the field showing complete invasion of the VNC in RasHi/Lkb1 vs. no invasion for RasHi discs.

The same lack of number of observations and a control is true of the results presented in Figure 4 and the authors should discuss the limitation of their observations. I am not sure what this aspect of the study adds to the field in its current preliminary state.

Reviewer #1 is absolutely correct. We have removed the data and conclusions presented in the original Figure 4, which was not central to the main findings presented in the manuscript.

Minor specific items to address to make paper clearer to reader and strengthen the conclusions made from the data:

1. The two UAS-RasV12 lines used and characterized as either Hi or Lo need to be appropriately referenced – only RasLo is referenced and RasHi is named as the same line in Materials and methods. Are they different strains? Catalogue numbers for Bloomington?

We have appropriately referenced both RasHi and RasLo stocks in the Materials and Methods.

2. In line 135 the authors should state they assayed for CLEAVED death caspase 1 in mutant clones to specify activation of cell death pathway.

We have corrected this important oversight.

3. In Figure 2A the authors show examples of third instar eye imaginal discs and claim that the RasLo Lkb1^{-/-} tissue is larger but there is less GFP⁺ mutant clonal tissue. The represented examples in the figure do support this however in Figure 1 panel D the difference between the control and RasLo Lkb1^{-/-} does not seem very evident. Is this a good representative image?

Thank you for pointing this out. We have added confocal images of an appropriately aged disc that represents the hyperplastic phenotype of RasLo Lkb1^{-/-} tissue.

4. In line 161 the authors state that altogether these data suggest 'they undergo G1 arrest while promoting the increased hyperplastic proliferation of surrounding WT tissue'. I believe this conclusion is correct and the FACS data in Supplemental Figure 2 is the evidence that supports it and so that data should be explained and referenced in this paragraph. Currently only the BrdU analysis is mentioned.

Since the data in former Suppl. Fig. 2 is central to the findings of the manuscript, we have added the data to what is now Figure 2, and have explained and referenced the FACS data in the accompanying Results section.

5. Where two examples are given of the representative clonal tissue of Ras^{Hi} Lkb1^{-/-} tissue (Figure 2B), the authors should explain in the figure legend what 13d AEL means – I assume this is an example of the imaginal discs of the giant larvae at day 13 having failed to pupate.

Reviewer #1 is correct and the stage '13d AEL' has been appropriately explained in the figure legend.

6. The survival analysis following transplantation of tumour tissue in Figure 2G shows a significant survival disadvantage of Ras^{Lo} Lkb1^{-/-} larvae as well but this is not discussed. The GFP levels do look a lot lower – is the tumour tissue still alive, growing, migrating? It should be acknowledged/discussed in light of the elevated MPP levels seen by IF in Figure 2E.

We have addressed this in the text by postulating that we cannot rule out the evolution and potential partial transformation of the overgrown GFP- tissue from Ras^{Lo}/Lkb1^{-/-} discs.

7. The pAmpk western in Figure 5C is not very convincing – if this has been done 3 times is this the best result achieved? The IF of pAMPK (5B) does support the conclusion but due to the unconvincing Western I think a Ras^{Hi} Lkb1^{+/+} control is needed for the IF to claim this is specifically due to the absence of Lkb1. I think either stronger data needs to be presented or the wording in the text should honestly reflect what is shown.

Yes, despite repeating the pAmpk western multiple times using different conditions, this is the clearest band we can achieve. One aspect of the biology of these tumors is that they are full of acidic vesicles we believe to be autolysosomes. In fact, the tumors are so acidic that by day 18 AEL the GFP+ tumors have photoconverted and can be visualized using our RFP filters. Since Lkb1/Ampk signaling has been shown to occur on endosomes (O'Farrell et al. 2017), it is possible that the quality of the band is being affected by cell compartment and/or pH. We believe that the western, combined with the IF and genetic data together provide strong evidence that Ampk remains active in Ras^{Hi}/Lkb1^{-/-} tumors.

8. Is the Ampk RNAi fly validated? If so reference, if not use the pAmpk antibody to show successful knockdown in clones by either western or IF.

We have validated the percent knockdown of the ampk^{Trip20} line in our hands (see Suppl. Fig. 2)

9. Is KN-93 a well validated compound with on target effects or in clinical trials? Include references to support its use as a specific CAMKK inhibitor. If not again use western to validate on target effect on pathway after feeding larvae.

KN-93 is a well-validated inhibitor of multiple members of the CamK pathway. We have referenced an extensive review in the text to support its use. Since the sole Drosophila ortholog has conserved features of multiple CaMKs we thought this was the best strategy to capture 'proof-of-principle' data. Lastly, we have validated the compound in our hands using western analysis of tumors from KN-93-fed Ras^{H1}/Lkb1^{-/-} larvae (see Suppl. Fig. 2).

10. CCAA identified three sub circuits of the AMPK pathway upregulated in RasHi Lkb1^{-/-} compared to RasHi although the relevance of these to cancer was not clear. Are they targetable therapeutically?

We have updated the text to include references on the cancer-relevance of each upregulated circuit/node.

I believe addressing the above comments will improve the paper but the major conclusions come from the molecular studies of Figure 5 and translating the finding to relevant human cancer patient cohorts. The conclusions of Figure 5 are that loss of Lkb1 in combination with high levels of oncogenic Ras drive co-activation of both mTOR and AMPK pathway however only AMPK pathway was followed up. The authors should extend their studies to include the contribution of mTOR in vivo or analyse it in terms of the human patient cohort data.

LKB1 is a well-established negative regulator of mTOR signaling (reviewed in Momcilovic and Shackelford, 2015); however, single-agent rapalogs have shown no therapeutic benefit in Kras/Lkb1 lung cancer GEMMs, and have performed poorly in lung cancer clinical trials. For these reasons, we initially decided not to follow-up on the mTOR pathway. We have updated the conclusions of Figure 5 to focus solely on the CaMK/Ampk signaling pathway – which to our knowledge has yet to be pursued as a therapeutic vulnerability in Kras/Lkb1-mutant lung cancer.

I think to conclude with the claim that the AMPK pathway is important and is activated through CamK to induce autophagy for a pro-tumourigenic effect requires strengthening of the data in Figure 5 with further western blots, genetic or pharmacological interventions eg. Rapamycin to block mTORC1 (DONE), metformin

to block AMPK (This has already been published in mammalian models). Perhaps a schematic model of the pathway would help visualise how the authors believe the tumours are 're-wired' following loss of Lkb1 and high levels of oncogenic Ras to promote tumour growth.

Since rapamycin has not performed well in pre-clinical or clinical studies as monotherapy, and metformin is controversial due to its indirect effect on Ampk, we have removed the conclusion that $Kras^{Hi}/Lkb1^{-/-}$ tumors are 're-wired' metabolically. The lipidated Atg8 western blot remains in the manuscript to add further evidence that Ampk is active in these tumors.

The human lung patient data from the TCGA presented in Figure 6 supports their stratification of high and low KRAS levels and I realise both KRAS and LKB1 are highly mutated in lung cancer. I just wonder if the authors have looked for the association in any other type of cancer eg. Colon cancer - LKB1 important tumour suppressor and Pancreatic cancer - KRAS mutation major driver. This could further extend the interest of the study to researchers focused on other cancers and potentially open up new treatment targets for more patients.

We did investigate this. Since there is a paucity of co-mutated KRAS/LKB1-mutant tumors outside of lung cancer, we combined the data from pancreatic, colorectal, melanoma, uterine, and bladder cancer but saw no survival differences based on stratification of KRAS levels or copy number. The phenomenon is specific to lung cancer – and seems to be most pronounced for patients with KRAS codon 12 mutations (which of course are predominant in lung cancer).

Reviewer #2 (Remarks to the Author):

Minor changes:

Fig 2d: scale bar is misplaced

Fixed in what is now Figure 3c

Fig 3d SD is missing

We have included the S.E.M. in what is now Figure 4b

Fig 5a control loading is missing

The growth pathways and the CamKII and lipidated Atg8 western were all probed on the same membrane. They have been combined into one panel with the relevant Actin control blot.

Fig 5a the phospho-residues recognized by the antibodies should be written in the figure and the same is for 5c and d.

Done.

Fig 5e add the quantification of GFP and statistics.

We scaled up and independently repeated the ampk genetic rescue experiment three times. From our new expanded analysis, it became clear that the predominant phenotype of $Kras^{Hi}/Lkb1^{-/-}$ rescued adult flies was hyperplastic eyes with no GFP signal. For quantitation, we have calculated the percent adult viability (now Fig. 5d,e).

Fig 6 quality of image g must improve the words are not clearly visible at least from the printout submitted

We have improved the quality of the image in Fig. 6g to highlight the relevant signaling nodes.